# Contrastive Weak-to-strong Generalization

Houcheng Jiang [1 2]   Junfeng Fang [3]   Jiaxin Wu [1]   Tianyu Zhang [1]
Chen Gao [4 2]   Xiang Wang [1 *]   Xiangnan He [1 *]   Yang Deng [5]

## Abstract

Weak-to-strong generalization provides a promising paradigm for scaling large language models (LLMs) by training stronger models on samples from aligned weaker ones, without requiring human feedback or explicit reward modeling. However, its robustness and generalization are hindered by the noise and biases in weak-model outputs, which limit its applicability in practice. To address this challenge, we leverage implicit rewards, which approximate explicit rewards through log-likelihood ratios, and reveal their structural equivalence with Contrastive Decoding (CD), a decoding strategy shown to reduce noise in LLM generation. Building on this connection, we propose **Contrastive Weak-to-Strong Generalization (ConG)**, a framework that employs contrastive decoding between pre- and post-alignment weak models to generate higher-quality samples. This approach enables more reliable capability transfer, denoising, and improved robustness, substantially mitigating the limitations of traditional weak-to-strong methods. Empirical results across different model families confirm consistent improvements, demonstrating the generality and effectiveness of ConG. Taken together, our findings highlight the potential of ConG to advance weak-to-strong generalization and provide a promising pathway toward AGI. Our code is available at: https://github.com/jianghoucheng/ConG

## 1. Introduction

Weak-to-strong generalization has emerged as a promising paradigm for scaling the capabilities of large language models (LLMs) (Burns et al., 2024; Yao et al., 2025; Li et al., 2025; Somerstep et al., 2025; Zhou et al., 2024b). By leveraging supervised samples generated from an aligned weaker model, a stronger model can be directly trained without requiring additional reward modeling or human feedback (Burns et al., 2024; Ouyang et al., 2022; Lee et al., 2024). This enables LLMs to transfer and extend capabilities to even stronger models, providing opportunities for self-enhancement and thus offering a potential pathway toward Artificial General Intelligence (AGI) (Goertzel, 2014).

Despite encouraging progress, the paradigm suffers from poor robustness and limited generalization (Yao et al., 2025; Yang et al., 2025). We attribute this to the inherent biases and preferences embedded in the weaker model: the samples it generates often contain noise and are of relatively low quality. As a result, the stronger model fails to generalize reliably, thereby restricting the applicability of weak-to-strong methods (Lyu et al., 2025). This raises a central research question: *How can we extract higher-quality samples from weak models, without relying on explicit rewards (e.g., human feedback or reward models), to achieve more effective weak-to-strong generalization?*

Recent success of implicit rewards in preference alignment and reasoning enhancement motivates our approach (Yuan et al., 2024; Cui et al., 2025). Specifically, implicit reward parameterizes the reward as the log-likelihood ratio between outputs from the post-alignment and pre-alignment models, and prior work has shown it to be an unbiased approximation of explicit reward (Rafailov et al., 2023). This suggests that implicit reward can serve as a reliable signal for assessing sample quality. Moreover, we observe that its log-ratio structure closely matches the form of **Contrastive Decoding (CD)**, a recently proposed decoding strategy proven to mitigate noise in LLM generation (Li et al., 2023). This structural consistency implies that the **CD process can be interpreted as generating responses that maximize implicit reward**. We formalize this conclusion as the *CD–Implicit Reward Correlation*. Empirically, Figure 1(b) shows a strong linear correlation between implicit

[1]University of Science and Technology of China [2]Zhongguancun Academy [3]National University of Singapore [4]Tsinghua University [5]Singapore Management University. Correspondence to: Xiang Wang < xiangwang1223@gmail.com>, Xiangnan He < xiangnanhe@gmail.com>.

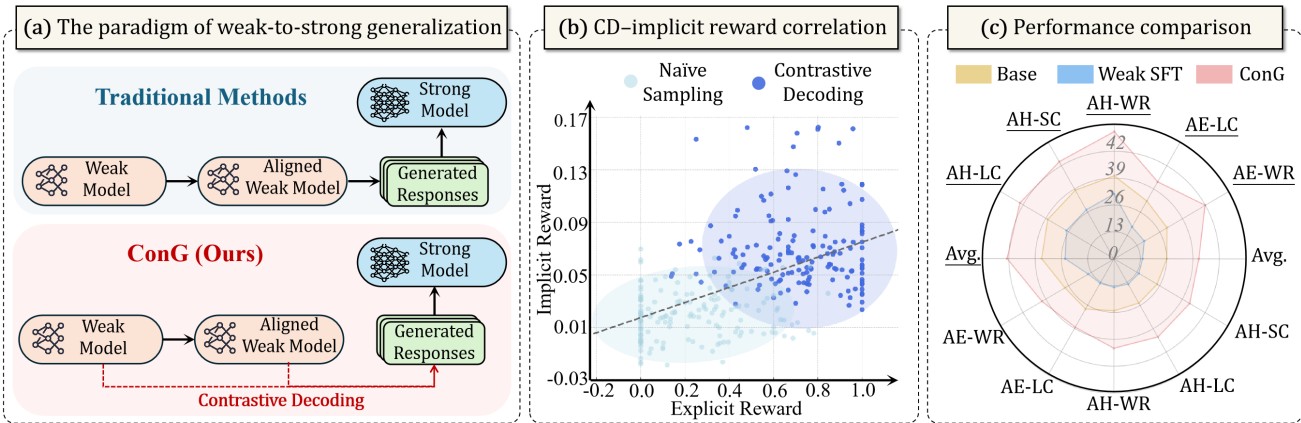

*Figure 1.* Overview of our proposed ConG. (a) Paradigm illustration comparing traditional weak-to-strong methods with ConG. (b) Scatter plot showing the correlation between implicit and explicit rewards, together with a comparison of sample rewards from naive sampling and contrastive decoding. (c) Radar chart comparing weak-to-strong methods on AlpacaEval2 (AE) and Arena-Hard (AH); metrics with underlines denote Qwen2.5-7B-Instruct, while those without underlines correspond to Llama3-8B-Instruct. Best viewed in color.

and explicit rewards, with CD-generated samples concentrating in the high implicit-reward region, consistent with this correlation.

The CD–Implicit Reward Correlation provides a practical pathway for generating higher-quality samples from weak models to train stronger ones, enabling weak-to-strong generalization with reduced noise risk. As illustrated in Figure 1(a), instead of directly using samples generated by the aligned weak model, we employ contrastive decoding between the pre-alignment and post-alignment weak models to generate training samples for the strong model. Theoretically, samples obtained via contrastive decoding preserve the full signal of target preferences and approximately maximize implicit reward, thereby supporting effective weak-to-strong generalization. We refer to this simple yet effect paradigm as **Contrastive Weak-to-Strong Generalization (ConG)**. In addition to weak-to-strong generalization, ConG can naturally reduce to *self-alignment* when the weak and strong models are instantiated as the same model, further broadening its applicability.

We evaluate our approach on two mainstream LLM families—Qwen2.5 (Yang et al., 2024) and Llama3 (Dubey et al., 2024). All models are trained for both weak-to-strong alignment and self-alignment using the UltraFeedback dataset (Cui et al., 2024), and evaluated on AlpacaEval2 (Dubois et al., 2024) and Arena-Hard (Li et al., 2024). Experimental results demonstrate that ConG consistently and significantly outperforms traditional weak-to-strong methods across all models. On average, it yields a gain of about 16.5% over the base model, as shown in Figure 1(c). These results confirm the generality of ConG across different alignment scenarios and highlight its ability to improve capability transfer, denoising, and robustness, offering a promising pathway toward AGI.

## 2. Preliminary

### 2.1. Reinforcement Learning from Human Feedback

In preference alignment, the training dataset $\mathcal{D} = \{(x, y_w, y_l)\}$ consists of prompts $x$ paired with two candidate responses, where $y_w \succ y_l \mid x$ indicates that $y_w$ is preferred over $y_l$. A common assumption is that preferences follow the Bradley–Terry model (Bradley & Terry, 1952), in which the probability of preferring $y_w$ over $y_l$ is proportional to the exponentiated reward. Under this assumption, a reward model $r_\theta(x, y)$ is trained by maximizing the pairwise log-likelihood:

$$\mathcal{L}_{\text{RM}}(r_\theta) = -\mathbb{E}_{(x,y_w,y_l)\sim\mathcal{D}} \left[\log \sigma\left(r_\theta(x, y_w) - r_\theta(x, y_l)\right)\right], \tag{1}$$

where $\sigma(\cdot)$ denotes the sigmoid function. Once the reward model is trained, the language model is treated as a policy $\pi_\theta(y \mid x)$ and optimized to maximize the expected reward while remaining close to a reference policy $\pi_{\text{ref}}$:

$$\max_{\pi_\theta} \mathbb{E}_{x\sim\mathcal{D},y\sim\pi_\theta(\cdot|x)}[r_\theta(x, y)] - \beta\text{KL}\left(\pi_\theta(\cdot|x)\|\pi_{\text{ref}}(\cdot|x)\right), \tag{2}$$

where $\beta > 0$ controls the regularization strength. This objective is typically optimized using Proximal Policy Optimization (PPO) (Schulman et al., 2017; Ouyang et al., 2022; Christiano et al., 2017).

### 2.2. Direct Preference Optimization (DPO)

Direct Preference Optimization (DPO) (Rafailov et al., 2023) provides a stable, scalable alternative to RLHF by directly optimizing the policy from preference comparisons, without an explicit reward model or on-policy sampling. DPO can be derived by reparameterizing the reward function $r(x, y)$ through the closed-form solution to the KL-

regularized reward maximization problem:

$$r(x, y) = \beta \log \frac{\pi_r(y \mid x)}{\pi_{\text{ref}}(y \mid x)} + \beta \log Z(x), \quad (3)$$

where $\pi_r$ is the aligned policy, $\pi_{\text{ref}}$ is the reference policy, and $Z(x)$ is a partition function independent of $y$. The corresponding *implicit reward* for a trainable policy $\pi_\theta$ is

$$\hat{r}(x, y) = \beta \log \frac{\pi_\theta(y \mid x)}{\pi_{\text{ref}}(y \mid x)}. \quad (4)$$

Given the preference dataset $\mathcal{D} = \{(x, y_w, y_l)\}$ (with $y_w \succ y_l \mid x$), substituting $\hat{r}$ into the pairwise likelihood yields the DPO loss:

$$\mathcal{L}_{\text{DPO}}(\pi_\theta) = -\mathbb{E}_{(x, y_w, y_l) \sim \mathcal{D}} \Big[ \log \sigma\big(\hat{r}(x, y_w) - \hat{r}(x, y_l)\big) \Big]. \quad (5)$$

# 3. CD–Implicit Reward Correlation

In this section, we begin by discussing how language models can be interpreted as implicit rewards that capture preferences (Section 3.1). Building on this foundation, Section 3.2 demonstrates that the formulation of implicit rewards is mathematically consistent with contrastive decoding, leading to the formalization of the CD–Implicit Reward Correlation. Finally, Section 3.3 presents empirical evidence showing that contrastive decoding outputs inherently encode preference information and provide higher-quality supervision signals, thereby laying the foundation for the weak-to-strong generalization method introduced in the next section.

## 3.1. Language Models as Reward Functions

As defined in Eqn. 4, the implicit reward $\hat{r}(x, y)$ can be expressed as the difference in log-probabilities between the aligned policy $\pi_r$ and the reference policy $\pi_{\text{ref}}$. By factorizing the log-probability at the token level (Zhou et al., 2024b), we obtain:

$$\hat{r}(x, y) = \beta \sum_{t=1}^{|y|} \big[\log \pi_\theta(y_t \mid x, y_{<t}) - \log \pi_{\text{ref}}(y_t \mid x, y_{<t})\big], \quad (6)$$

where $y_{<t}$ denotes the prefix of the first $t - 1$ tokens of $y$, and $y_t$ denotes the token at position $t$. This token-level decomposition shows that the implicit reward can be viewed as the sum of per-token log-probability differences between the aligned policy $\pi_r$ and the reference policy $\pi_{\text{ref}}$. In practice, this token-level implicit reward serves as a dense, model-based proxy for the explicit reward $r(x, y)$ provided by a trained reward model, enabling preference estimation directly from model behavior without additional annotation.

## 3.2. Connection between Contrastive Decoding and Implicit Reward

As shown in Section 3.1, the token-level implicit reward measures the relative preference between two policies $\pi_r$ and $\pi_{\text{ref}}$ via the per-token log-probability difference. This formulation is mathematically consistent with the inference-time decoding strategy–contrastive decoding (Li et al., 2023), which generates tokens by comparing their probability distribution under $\pi_r$ and $\pi_{\text{ref}}$.

At each decoding step $t$, contrastive decoding defines the next-token distribution as:

$$p_\alpha(y_t \mid x, y_{<t}) = \text{softmax}\big(\mathcal{F}(y_t)\big),$$
$$\mathcal{F}(y_t) = \begin{cases} g(y_t), & \text{if } y_t \in \mathcal{V}_{\text{head}} \\ -\infty, & \text{otherwise} \end{cases}$$
$$\text{where } g(y_t) = (1 - \alpha) \log \frac{\pi_r(y_t \mid x, y_{<t})}{\pi_{\text{ref}}(y_t \mid x, y_{<t})}$$
$$+ \alpha \log \pi_r(y_t \mid x, y_{<t}). \quad (7)$$

where $\alpha \in [0, 1]$ (contrastive coefficient) controls the relative weight of the contrastive term. Smaller $\alpha$ increases the influence of the probability gap, biasing the model toward higher-implicit-reward tokens, while $\alpha = 1$ reduces to standard decoding under $\pi_r$.

The candidate set $\mathcal{V}_{\text{head}}$ is obtained via vocabulary pruning:

$$\mathcal{V}_{\text{head}}(x, y_{<t}) = \big\{ y_t \in \mathcal{X} : \pi_r(y_t \mid x, y_{<t}) \geq \lambda \cdot \max_w \pi_r(w \mid x, y_{<t})\big\}, \quad (8)$$

where $\lambda \in [0, 1]$ is the threshold. If the predicted probability of a token under $\pi_r$ is far smaller than the top candidate in the same decoding step, it is unlikely to be a reasonable prediction; thus, such tokens are excluded from $\mathcal{V}_{\text{head}}$ by setting their logits to $-\infty$ in Eqn. 7.

Comparing Eqn. 7 with the token-level implicit reward in Eqn. 4 shows that the contrastive term $\log \frac{\pi_r(y_t \mid x, y_{<t})}{\pi_{\text{ref}}(y_t \mid x, y_{<t})}$ is exactly the implicit reward up to a scaling factor $(1 - \alpha)$. Therefore, under the contrastive decoding distribution $p_\alpha$, the decoding objective is to find $y^*$ that approximately maximizes the implicit reward, as follows:

$$y^* = \arg\max_y \sum_{t=1}^{|y|} \Big[ (1 - \alpha) \log \frac{\pi_r(y_t \mid x, y_{<t})}{\pi_{\text{ref}}(y_t \mid x, y_{<t})}$$
$$+ \alpha \log \pi_r(y_t \mid x, y_{<t}) \Big]. \quad (9)$$

Based on this derivation, samples generated with contrastive decoding are expected to have higher implicit rewards than standard decoding, and $\alpha$ offers a direct control over the implicit reward level by adjusting the weight of the contrastive term (see Appendix C for a detailed proof). We formalize this conclusion as the *CD–Implicit Reward Correlation*.

### 3.3. Empirical Analysis

In Section 3.2, we established the theoretical correspondence between contrastive decoding and implicit rewards. We empirically examine this connection by varying the contrastive coefficient $\alpha$ and analyzing its effect on both rewards and generation behavior.

We first consider how $\alpha$ influences the implicit and explicit rewards. The results, summarized in Figure 2 (a) and Figure 2 (d), reveal a clear pattern: implicit reward decreases monotonically with increasing $\alpha$, consistent with our theoretical prediction, while explicit reward remains relatively high for $\alpha \in [0, 0.5]$ before dropping sharply once $\alpha > 0.5$. This suggests that smaller $\alpha$ values preserve more preference information from $\pi_r$, whereas larger values overemphasize the contrastive term and deteriorate overall generation quality.

To ensure that these differences are not confounded by superficial factors such as response length, we analyze the length distributions across $\alpha$ in Figure 2 (b) and Figure 2 (e). The distributions remain stable regardless of $\alpha$, indicating that the observed performance variation arises from content quality rather than response length bias.

Finally, the relative performance across different $\alpha$ settings is captured by the win-rate matrices in Figure 2 (c) and Figure 2 (f). Generations with $\alpha \in [0, 0.5]$ consistently outperform those with $\alpha \in [0.6, 1.0]$, reinforcing the observation that moderate contrastive strength leads to higher-quality outputs. Taken together, these results provide strong empirical support for our theoretical analysis, highlighting the critical role of $\alpha$ in balancing reward alignment and generation robustness.

## 4. Contrastive Weak-to-Strong Generalization (ConG)

Building on the CD–Implicit Reward Correlation established in Section 3.3, we introduce Contrastive Weak-to-Strong Generalization (ConG). The central idea is to leverage contrastive decoding (CD) to extract higher-quality responses from weak models, and use them to drive the generalization of stronger models. ConG consists of two stages: (i) ConG-S, which employs CD responses for SFT to provide a high-reward initialization, and (ii) ConG, which further strengthens weak-to-strong generalization with DPO.

**Stage I: ConG-S (Contrastive Decoding for SFT).** Let $\pi_r^w$ denote the post-alignment weak model and $\pi_{\text{ref}}^w$ the pre-alignment weak model. For each prompt $x \in \mathcal{X}$, we instantiate contrastive decoding with the weak models to obtain the decoding distribution defined in Eqn. 8. Decoding under $p_\alpha$ yields a chosen sample $y_w$ for each $x$, forming $\mathcal{D}_{\text{SFT}} = \{(x, y_w)\}$. Let $\pi_{\text{ref}}^s$ be the initial strong model; we

obtain $\pi_{\text{SFT}}^s$ by minimizing the standard SFT loss on chosen samples:

$$\mathcal{L}_{\text{SFT}}(\pi_\theta^s) = -\mathbb{E}_{(x,y_w)\sim\mathcal{D}_{\text{SFT}}}\left[\log \pi_\theta^s(y_w \mid x)\right]. \quad (10)$$

This stage moves the strong model's policy toward the CD–induced preference distribution, providing a high-reward starting point for preference optimization.

**Stage II: ConG (Generalization with DPO).** After SFT, we further refine the strong model using DPO. For each prompt $x$, we sample an additional response $y_l$ from $\pi_{\text{SFT}}^s$ under standard decoding and pair it with the corresponding CD response $y_w$. This construction is justified by two factors: (i) CD responses approximate maximizers of implicit reward (Eqn. 9), so in expectation their implicit reward satisfies $\hat{r}_w > \hat{r}_l$ (see Appendix C for a detailed proof). (ii) since $\pi_{\text{SFT}}^s$ has been trained on $y_w$, the distributions of $y_w$ and $y_l$ are well matched, ensuring that their comparison isolates reward differences rather than distributional shifts. We then optimize the strong model with the DPO loss, taking $\pi_{\text{SFT}}^s$ as the reference policy:

$$\mathcal{L}_{\text{DPO}}(\pi_\theta^s) = -\mathbb{E}_{(x,y_w,y_l)\sim\mathcal{D}_{\text{DPO}}}\left[\log \sigma \left(\vphantom{\frac{\pi_\theta^s}{\pi_{\text{SFT}}^s}}\right.\right.$$
$$\left.\left. \beta \log \frac{\pi_\theta^s(y_w \mid x)}{\pi_{\text{SFT}}^s(y_w \mid x)} - \beta \log \frac{\pi_\theta^s(y_l \mid x)}{\pi_{\text{SFT}}^s(y_l \mid x)}\right)\right] \quad (11)$$

where $\beta > 0$ controls preference sharpness and $\sigma(\cdot)$ denotes the logistic function. This stage exploits the reward gap $\hat{r}_w > \hat{r}_l$ while maintaining distributional consistency, thereby pushing the strong model toward more reliable and robust generalization.

Together, ConG-S and ConG constitute our framework for contrastive weak-to-strong generalization. A detailed algorithmic description is provided in Appendix A. Notably, when $\pi_{\text{ref}}^w = \pi_{\text{ref}}^s$, ConG naturally reduces to a form of *self-alignment*.

## 5. Experiments

In this section, we conduct extensive experiments to address the following research questions:

- **RQ1:** How does our proposed ConG-S and ConG perform compared to baseline approaches in both weak-to-strong alignment and self-alignment settings?

- **RQ2:** How does the effectiveness of ConG-S and ConG in weak-to-strong alignment vary with different capability gaps between the weak and strong models?

- **RQ3:** How does the contrastive coefficient $\alpha$ influence the performance of ConG-S and ConG?

- **RQ4:** How do ConG-S and ConG affect downstream evaluations, and to what extent do they preserve the model's general capabilities without degradation?

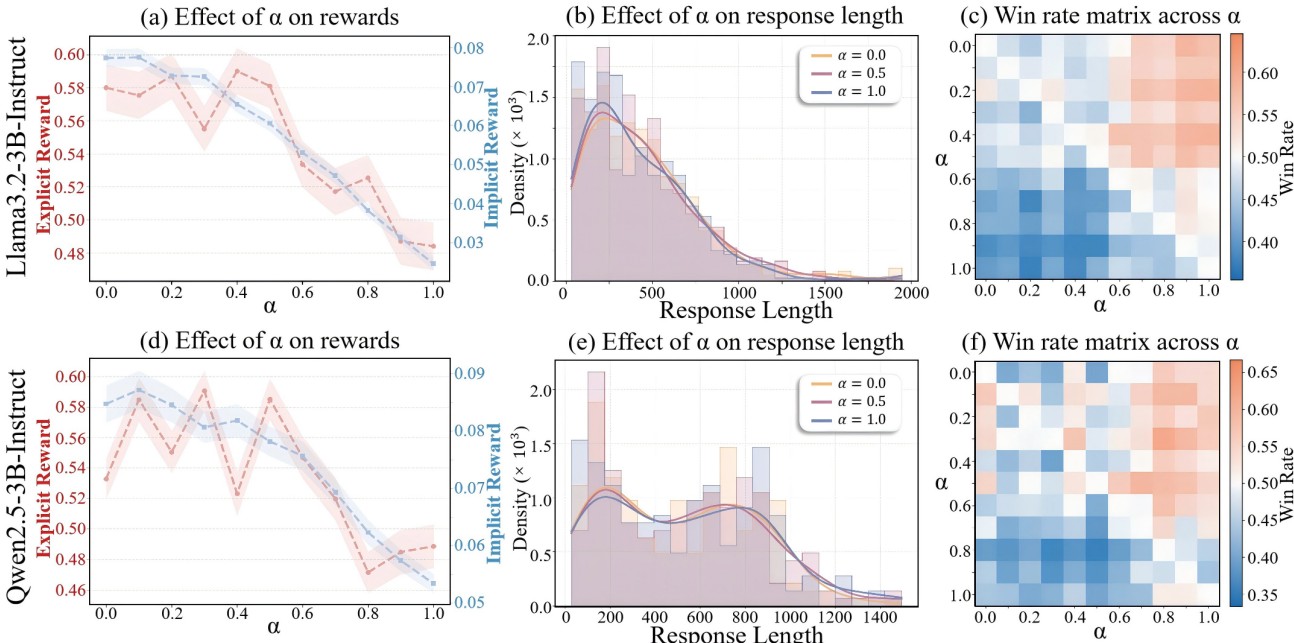

*Figure 2.* Performance comparison of contrastive decoding with different contrastive coefficients $\alpha$ for Llama3.2-3B-Instruct (top row) and Qwen2.5-3B-Instruct (bottom row). (a,d) Relationship between $\alpha$ and implicit/explicit rewards. (b,e) Relationship between $\alpha$ and response length. (c,f) Win-rate matrices showing the proportion of cases where row $\alpha$ outperforms column $\alpha$.

## 5.1. Experimental Setup

In this subsection, we summarize the base models, dataset, benchmarks and baselines used in our experiments. Further details and additional experiments are provided in Appendix B and D.

**Base Models and Dataset.** We conduct experiments on two widely used model families: Qwen2.5 (Yang et al., 2024) and Llama3 (Dubey et al., 2024). For weak-to-strong alignment, we follow prior works (Zhu et al., 2025; Lyu et al., 2025) and select the smaller-parameter models within each family, Qwen2.5-3B-Instruct and Llama3.2-3B-Instruct, as the reference weak models $\pi_{\text{ref}}^{\text{weak}}$, and the larger-parameter counterparts, Qwen2.5-7B-Instruct and Llama3-8B-Instruct, as the reference strong models $\pi_{\text{ref}}^{\text{strong}}$. We further construct the aligned weak model $\pi_{r}^{\text{weak}}$ by fine-tuning each reference weak model with DPO, which is then used to facilitate contrastive decoding and provide chosen samples. For preference alignment data, we adopt the UltraFeedback dataset (Cui et al., 2024). To construct pairwise supervision for DPO training, to rank the self-generated samples from the reference weak models and generate pairwise annotations.

**Evaluation Benchmarks and Baselines.** We evaluate our method on two widely used instruction-following benchmarks, AlpacaEval2 (Dubois et al., 2024) and Arena-Hard (Li et al., 2024). For AlpacaEval2, we report both the length-controlled win rate (LC) and the raw win rate (WR). For Arena-Hard, we report the win rate (WR) and the style-controlled win rate (SC). For baselines, we adopt different experimental settings for weak and strong models. For weak models, we evaluate our method in the self-alignment setting (ConG-S *(self)* and ConG *(self)*) against preference optimization algorithms, including DPO (Rafailov et al., 2023), ORPO (Hong et al., 2024), and SimPO (Meng et al., 2024). For strong models, we focus on the weak-to-strong alignment setting, where supervision signals must come from the weak model for fairness. We therefore evaluate our method (ConG-S *(w→s)* and ConG *(w→s)*) against established weak-to-strong approaches: Weak SFT (pre) (Burns et al., 2024), which uses the pre-alignment weak model; Weak SFT (post), which instead uses the post-alignment weak model; WeakTeacher (Tao & Li, 2025); AuxConf (Burns et al., 2024), a variant of Weak SFT with an auxiliary confidence loss; and WSPO (Zhu et al., 2025).

## 5.2. Weak-to-Strong Alignment Results (RQ1)

We first evaluate the effectiveness of our method in both the self-alignment and weak-to-strong alignment settings. Specifically, we use Qwen2.5-3B-Instruct and Llama3.2-3B-Instruct as weak models to improve the performance of the Qwen2.5-7B-Instruct and Llama3-8B-Instruct strong models, respectively. The results are summarized in Table 1, from which we find that:

- **Obs 1: In the self-alignment setting, our method significantly improves over existing preference optimiza-**

*Table 1.* **Main results on AlpacaEval2 and Arena-Hard.** "Base" denotes the reference model without additional alignment. "WR" denotes the raw win rate, "LC" the length-controlled win rate, "SC" the style-controlled win rate, and "Avg." the average score across benchmarks. Best results are highlighted in bold and second-best are underlined.

| | Qwen2.5-3B-Instruct (Weak) | | | | | Llama3.2-3B-Instruct (Weak) | | | | |
| | AlpacaEval 2 | | Arena-Hard | | Avg. | AlpacaEval 2 | | Arena-Hard | | Avg. |
| Method | LC | WR | SC | WR | | LC | WR | SC | WR | |
|---|---|---|---|---|---|---|---|---|---|---|
| Base | 13.8±0.3 | 14.3±1.5 | 30.5±2.3 | 33.8±2.7 | 22.8 | 20.2±0.4 | 23.8±1.4 | 22.6±2.2 | 20.2±2.6 | 21.8 |
| DPO | 29.4±0.4 | 34.9±1.5 | 42.4±2.5 | 44.3±2.7 | 37.8 | 31.4±0.3 | 34.8±1.7 | 29.8±2.1 | 29.3±2.4 | 29.6 |
| ORPO | 22.7±0.2 | 26.3±1.4 | 34.5±2.1 | 38.0±2.8 | 30.4 | 27.7±0.5 | 30.1±1.5 | 24.7±2.4 | 25.5±2.2 | 25.8 |
| SimPO | 34.1±0.3 | 35.9±1.3 | 45.7±2.8 | 50.1±2.9 | 41.5 | 34.0±0.4 | 36.9±1.4 | 32.0±2.5 | 31.2±2.3 | 33.5 |
| ConG-S *(self)* | 33.3±0.4 | 37.7±1.7 | 46.6±2.3 | 51.8±2.7 | 42.4 | 30.9±0.3 | 33.0±1.6 | 33.3±2.8 | 31.6±2.2 | 32.2 |
| ConG *(self)* | 35.9±0.5 | 43.3±1.6 | 49.2±2.4 | 53.5±2.9 | 45.5 | 34.7±0.2 | 37.8±1.3 | 33.3±2.5 | 32.6±2.8 | 34.6 |

| | Qwen2.5-7B-Instruct (Strong) | | | | | Llama3-8B-Instruct (Strong) | | | | |
| | AlpacaEval 2 | | Arena-Hard | | Avg. | AlpacaEval 2 | | Arena-Hard | | Avg. |
| Method | LC | WR | SC | WR | | LC | WR | SC | WR | |
|---|---|---|---|---|---|---|---|---|---|---|
| Base | 32.3±0.4 | 30.2±1.5 | 38.3±2.2 | 40.1±2.8 | 35.2 | 28.1±0.3 | 28.1±1.3 | 24.7±2.5 | 25.2±2.7 | 26.5 |
| Weak SFT (pre) | 17.8±0.3 | 17.2±1.4 | 27.4±2.6 | 31.3±2.7 | 23.4 | 13.7±0.4 | 14.8±1.6 | 14.1±2.8 | 13.8±2.5 | 14.1 |
| Weak SFT (post) | 33.0±0.3 | 31.0±1.3 | 38.9±2.4 | 40.7±2.7 | 35.9 | 28.7±0.3 | 28.8±1.5 | 25.3±2.6 | 25.9±2.5 | 27.2 |
| WeakTeacher | 36.8±0.4 | 38.7±1.6 | 46.5±2.5 | 52.0±2.8 | 43.5 | 30.8±0.4 | 32.1±1.6 | 33.5±2.7 | 34.0±2.8 | 32.6 |
| AuxConf | 21.2±0.2 | 20.7±1.3 | 27.2±2.4 | 32.1±2.6 | 25.3 | 16.7±0.5 | 19.5±1.7 | 14.3±2.3 | 13.5±2.6 | 16.0 |
| WSPO | 18.6±0.3 | 21.0±1.2 | 29.1±2.2 | 33.7±2.9 | 25.6 | 17.3±0.4 | 19.8±1.6 | 18.5±2.1 | 16.9±2.4 | 18.1 |
| ConG-S *(w→s)* | 38.7±0.5 | 43.1±1.7 | 52.4±2.5 | 57.8±2.9 | 48.0 | 33.7±0.2 | 34.3±1.4 | 39.6±2.3 | 39.2±2.8 | 36.7 |
| ConG *(w→s)* | 43.0±0.4 | 51.9±1.6 | 54.5±2.6 | 61.2±2.7 | 52.7 | 38.3±0.3 | 41.2±1.5 | 43.6±2.2 | 43.3±2.9 | 41.6 |

**tion baselines.** Specifically, ConG-S *(self)* and ConG *(self)* achieve average improvements of about **15.0%** and **17.8%**, respectively, compared to reference. This highlights that contrastive decoding outputs from models indeed provide higher-quality supervision signals.

- **Obs 2: In the weak-to-strong setting, our method achieves substantial improvements over the base models.** Specifically, ConG-S *(w→s)* and ConG *(w→s)* improve the average scores by around **11.5%** and **16.3%**, respectively. These results demonstrate that contrastive decoding effectively transfers preference information from weak to strong models, leading to stronger alignment. Notably, other baselines even degrade performance relative to the base models.

### 5.3. Effect of Model Capability Gap (RQ2)

From the results in Table 1, we observe that our method yields larger gains in self-alignment than in weak-to-strong alignment. This raises the question of whether the capability

gap between weak and strong models affects the improvement. Based on the experiments across different model transitions, the results in Figure 3 (a) and (b) lead to the following observations:

- **Obs 3: Smaller model capability gaps lead to larger improvements in weak-to-strong alignment.** Specifically, transitions where the weak and strong models have closer sizes (*e.g.,* 7B→7B) result in significantly higher alignment improvements compared to those with larger capability gaps (*e.g.,* 1.5B→7B). This indicates that when the weak model is closer in capacity to the strong model, the weak model can provide more effective supervision, leading to better alignment.

- **Obs 4: Larger strong models contribute to more substantial weak-to-strong alignment improvements.** Specifically, transitions involving stronger strong models (*e.g.,* 3B→7B) show significantly larger performance gains compared to transitions with smaller strong models (*e.g.,* 1.5B→3B). This suggests that larger strong models

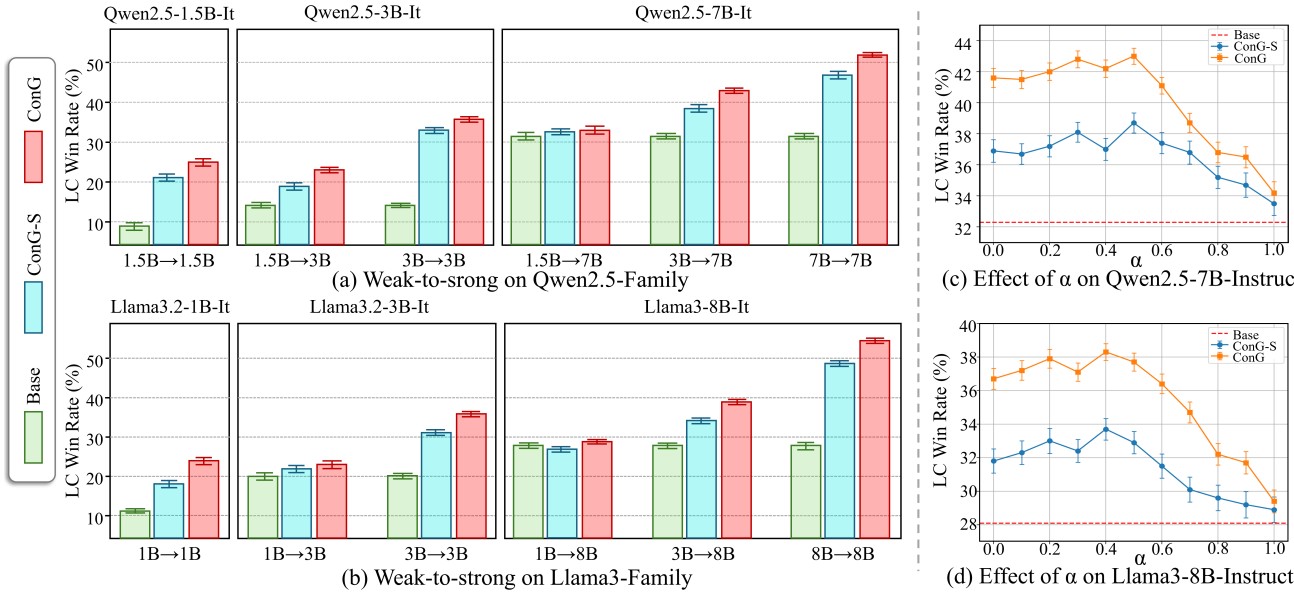

*Figure 3.* Performance of contrastive weak-to-strong generalization. (a) and (b) Results across different weak–strong model. (c) and (d) Effect of contrastive coefficient $\alpha$ on alignment performance, with error bars indicating standard errors. "Base" refers to the unaligned reference model.

are better at leveraging the preference signals from weak models, resulting in stronger alignment when trained on these signals.

### 5.4. Impact of $\alpha$ on Weak-to-Strong Alignment (RQ3)

We study the effect of the contrastive coefficient $\alpha$ on weak-to-strong alignment performance. As shown in Figure 3 (c) and (d), we report results across different $\alpha$ values. The figures show that:

- **Obs 5: Moderate values of $\alpha$ result in the best weak-to-strong alignment performance.** We find that setting $\alpha$ to moderate values (e.g., $\alpha = 0.3$ to $\alpha = 0.5$) consistently yields the highest performance gains. These values allow the contrastive term to significantly enhance the weak model's preference signal without overwhelming the original model behavior.

- **Obs 6: When $\alpha > 0.5$, performance improvement diminishes with increasing $\alpha$.** After $\alpha$ exceeds 0.5, further increases in $\alpha$ lead to diminishing returns in alignment performance. Specifically, as $\alpha$ grows larger, the additional benefits from the contrastive signal become less significant, and the overall improvement plateaus.

### 5.5. Downstream Task Evaluation (RQ4)

To assess whether ConG-based weak-to-strong generalization affects downstream task performance, we evaluate models trained with different alignment methods on a diverse

suite of benchmarks from the lm-eval-harness (Sutawika et al., 2024). The evaluation covers MMLU (Hendrycks et al., 2021), ARC (Clark et al., 2018), HellaSwag (Zellers et al., 2019), TruthfulQA (Lin et al., 2022), MathQA (Amini et al., 2019), and GSM8K (Cobbe et al., 2021). We follow standard evaluation protocols and report the results in Table 2, enabling a direct comparison of whether ConG preserves general capabilities. Based on these results, we draw the following observation:

- **Obs 7: ConG introduces negligible degradation on downstream tasks.** We observe that ConG and ConG-S maintain general-purpose capabilities nearly intact, with average changes within 1.0 points across benchmarks, indicating that our method preserves overall model utility.

## 6. Related Work

**AI alignment.** A central challenge in modern AI research is ensuring that advanced models act in line with human intent (Cao et al., 2024; Gao et al., 2024; Leike et al., 2018; Askell et al., 2021). Current alignment pipelines often depend on human feedback, most prominently RLHF (Ouyang et al., 2022; Christiano et al., 2017; Stiennon et al., 2020) and DPO (Rafailov et al., 2023). Recent studies have further extended alignment considerations to broader deployment settings, including safety-aware training and intervention mechanisms (Fang et al., 2026a; Jiang et al., 2026b). While effective at present, such approaches still scale poorly: human supervision becomes insufficient once models surpass expert ability, and collecting reliable annotations remains expensive and

*Table 2.* **Downstream task results on Qwen2.5-7B-Instruct and Llama3-8B-Instruct.** Differences relative to the Base are shown in parentheses. Improvements are marked in red, decreases in blue.

| Method | ARC_E | ARC_C | TruthfulQA | MMLU | MathQA | HellaSwag | GSM8K |
|---|---|---|---|---|---|---|---|
| **Qwen2.5-7B-Instruct** | | | | | | | |
| Base | 81.31 (+0.00) | 52.82 (+0.00) | 50.18 (+0.00) | 71.80 (+0.00) | 40.74 (+0.00) | 62.04 (+0.00) | 80.44 (+0.00) |
| Weak SFT | 76.02 (-5.29) | 50.01 (-2.81) | 47.12 (-3.06) | 67.55 (-4.25) | 38.12 (-2.62) | 61.44 (-0.60) | 75.28 (-5.16) |
| AuxConf | 77.81 (-3.50) | 51.37 (-1.45) | 49.24 (-0.94) | 68.12 (-3.68) | 41.02 (+0.28) | 60.85 (-1.19) | 76.42 (-4.02) |
| WSPO | 78.25 (-3.06) | 49.88 (-2.94) | 48.71 (-1.47) | 70.22 (-1.58) | 39.11 (-1.63) | 62.55 (+0.51) | 77.08 (-3.36) |
| ConG-S | 80.44 (-0.87) | 52.21 (-0.61) | 50.75 (+0.57) | 72.12 (+0.32) | 39.95 (-0.79) | 61.55 (-0.49) | 79.12 (-1.32) |
| ConG | 81.02 (-0.29) | 52.64 (-0.18) | 50.28 (+0.10) | 71.42 (-0.38) | 40.51 (-0.23) | 62.88 (+0.84) | 80.21 (-0.23) |
| **Llama3-8B-Instruct** | | | | | | | |
| Base | 81.40 (+0.00) | 52.99 (+0.00) | 46.76 (+0.00) | 63.81 (+0.00) | 42.01 (+0.00) | 57.71 (+0.00) | 75.28 (+0.00) |
| Weak SFT | 75.83 (-5.57) | 50.45 (-2.54) | 44.28 (-2.48) | 59.14 (-4.67) | 39.22 (-2.79) | 54.03 (-3.68) | 71.12 (-4.16) |
| AuxConf | 77.42 (-3.98) | 51.82 (-1.17) | 45.89 (-0.87) | 60.92 (-2.89) | 41.44 (-0.57) | 55.18 (-2.53) | 72.08 (-3.20) |
| WSPO | 78.04 (-3.36) | 49.91 (-3.08) | 46.42 (-0.34) | 62.08 (-1.73) | 40.36 (-1.65) | 58.02 (+0.31) | 73.21 (-2.07) |
| ConG-S | 80.62 (-0.78) | 52.05 (-0.94) | 47.19 (+0.43) | 64.01 (+0.20) | 41.12 (-0.89) | 57.25 (-0.46) | 74.38 (-0.90) |
| ConG | 81.22 (-0.18) | 52.71 (-0.28) | 46.93 (+0.17) | 63.54 (-0.27) | 42.41 (+0.40) | 57.38 (-0.33) | 75.66 (+0.38) |

difficult (Kim et al., 2024; Zeng et al., 2025). This challenge is also increasingly relevant beyond purely text-based models, as recent work explores alignment and reasoning in multimodal systems (Jiang et al., 2026a). These limitations motivate alignment strategies that reduce or bypass reliance on direct human oversight.

**Weak-to-strong generalization.** An alternative line of work explores whether stronger models can be trained under the supervision of weaker ones, a setting known as weak-to-strong generalization (W2SG). Initial evidence by (Burns et al., 2024) showed that strong models may outperform their weak teachers, pointing to the potential of this paradigm. Subsequent studies introduced new algorithms (Zhu et al., 2025; Lyu et al., 2025; Li et al., 2025; Somerstep et al., 2025; Zhou et al., 2024b; Mitchell et al., 2024; Zhou et al., 2024a; Tao & Li, 2025) and provided empirical analyses across tasks (Yao et al., 2025; Yang et al., 2025). Relatedly, distillation-based approaches also aim to transfer stronger behavioral signals into smaller or specialized models, including recent on-policy variants for language models and agents (Zhong et al., 2026; Fang et al., 2026b). Compared with these efforts, our work highlights a different perspective: we connect W2SG with implicit rewards and contrastive decoding, offering both conceptual justification and empirical validation on large language models.

**Contrastive Decoding.** Contrastive decoding lies at the core of our work. The key idea is to refine generation by contrasting probability distributions during decoding. The original contrastive decoding method (Li et al., 2023) contrasts a stronger expert model with a weaker amateur model to improve fluency and coherence. Subsequent extensions explore different contrastive dimensions: DoLa (Chuang et al., 2024) contrasts later, more mature layers with earlier layers to stabilize generation, while ICD (Zhang et al., 2023) contrasts models perturbed with hallucination-inducing noise to enhance factual accuracy. These studies highlight the flexibility of contrastive decoding and motivate our use of it as the foundation for W2SG.

# 7. Limitations and Future Works

Our work has several limitations. First, contrastive decoding is not yet fully compatible with mainstream inference acceleration techniques. Even though we adopt a pre-generation design to mitigate overhead, the additional computation still introduces latency compared to standard decoding, which may limit efficiency in practice. Second, ConG relies on maintaining multiple alignment states of the weak model to enable contrastive decoding. While the added complexity is moderate, it nevertheless increases the engineering burden relative to simpler pipelines.

To address these issues, we see several promising directions. A natural next step is to explore adapting contrastive decoding to fast inference paradigms such as speculative decoding and caching-based acceleration, which could greatly reduce latency. In addition, designing lighter-weight strategies to approximate multiple weak-model alignment states may simplify deployment without sacrificing effectiveness. We also plan to evaluate ConG in larger-scale, real-world applications, providing deeper insights into its practicality and robustness.

## 8. Conclusion

We presented **Contrastive Weak-to-Strong Generalization (ConG)**, a new paradigm that connects implicit rewards with contrastive decoding. Our key insight is that implicit rewards, parameterized as log-likelihood ratios, are structurally consistent with the mechanism of contrastive decoding. This correlation allows us to view contrastive decoding not only as a decoding strategy but also as a natural way of generating samples that maximize implicit reward. Building on this connection, ConG leverages contrastive decoding between aligned model states to provide higher-quality supervision signals, enabling more effective capability transfer, denoising, and improved robustness.

## Impact Statement

This work studies weak-to-strong generalization for LLMs and proposes a contrastive-decoding framework that reduces noise and bias in weak-model-generated training data without human feedback or explicit reward modeling, aiming to improve robustness and lower the cost of scalable alignment. The approach may benefit low-resource and domain-adaptation settings and help mitigate error amplification in training pipelines, but it can still inherit and propagate biases from the weak models if used without careful evaluation. We therefore emphasize responsible deployment, including dataset curation, testing across diverse settings, and complementary safety mechanisms.

## Acknowledgment

This research/project was supported by National Natural Science Foundation of China (U25A20445), the National Natural Science Foundation of China(62525211), the Zhongguancun Academy, and the Singapore Ministry of Education (MOE) Academic Research Fund (AcRF) Tier 1 grant (Proposal ID: 24-SIS-SMU-002). Yang Deng is support by the Lee Kong Chian Fellowship awarded by Singapore Management University.

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

# A. Algorithmic Details of ConG

We provide the unified algorithm for Contrastive Weak-to-Strong Generalization (ConG), which consists of two stages: (i) ConG-S, using contrastive decoding (CD) responses for SFT, and (ii) ConG, refining the strong model with DPO.

---

**Algorithm 1** Contrastive Weak-to-Strong Generalization (ConG)

---

**Input:** Pre-alignment weak model $\pi_{\text{ref}}^{\text{w}}$, post-alignment weak model $\pi_r^{\text{w}}$, initial strong model $\pi_{\text{ref}}^{\text{s}}$, prompt set $\mathcal{X}$, contrastive
     coefficient $\alpha$, preference sharpness $\beta$.
**Output:** Aligned strong model $\pi^{\text{s}}$.

**Stage I: ConG-S (Contrastive Decoding for SFT)**
  **for** *each prompt $x \in \mathcal{X}$* **do**
     Generate CD response $y_w$ from $(\pi_r^{\text{w}}, \pi_{\text{ref}}^{\text{w}}, \alpha)$.
     Add $(x, y_w)$ into dataset $\mathcal{D}_{\text{SFT}}$.

Train $\pi_{\text{ref}}^{\text{s}}$ on $\mathcal{D}_{\text{SFT}}$ with SFT loss (Eqn. 10) to obtain $\pi_{\text{SFT}}^{\text{s}}$.

**Stage II: ConG (Generalization with DPO)**
  **for** *each prompt $x \in \mathcal{X}$* **do**
     Retrieve $y_w$ from $\mathcal{D}_{\text{SFT}}$.
     Sample additional response $y_l \sim \pi_{\text{SFT}}^{\text{s}}$ with standard decoding.
     Construct pair $(y_w, y_l)$ with $r_w > r_l$.
     Add $(x, y_w, y_l)$ into dataset $\mathcal{D}_{\text{DPO}}$.

Train $\pi_{\text{SFT}}^{\text{s}}$ on $\mathcal{D}_{\text{DPO}}$ with DPO loss (Eqn. 11) to obtain final strong model $\pi^{\text{s}}$.

---

# B. Experimental Setup

## B.1. Benchmarks

Our evaluation is conducted on two widely adopted benchmarks for instruction-following language models: **AlpacaEval2** (Dubois et al., 2024) and **Arena-Hard** (Li et al., 2024). Both benchmarks follow a pairwise comparison paradigm, in which the output of a tested model is directly compared against that of a reference model on a shared prompt, and a stronger external judge model decides which response better satisfies the instruction. This setup provides a scalable and reliable approximation of human preference judgments, while controlling for common biases such as verbosity and stylistic artifacts. Below we provide detailed descriptions of these two benchmarks, the evaluation methodology, and the specific settings used in our experiments.

**AlpacaEval2.** AlpacaEval2 is the successor of AlpacaEval, designed to address issues of bias and instability in preference-based evaluation. It consists of around 805 diverse prompts that span open-ended question answering, reasoning, creative writing, and general instruction-following tasks. The evaluation protocol compares model responses head-to-head with those of a reference baseline model, with each pair adjudicated by a strong LLM judge. The main metric is the raw *win rate (WR)*, defined as the proportion of cases where the evaluated model's response is preferred over the baseline's. To reduce artifacts, AlpacaEval2 additionally reports the *length-controlled win rate (LC)*: a variant that normalizes for verbosity bias, since longer responses often receive higher preference regardless of actual quality. Length control is performed by conditioning on response length differences and recalibrating the win rate, thereby offering a fairer view of content quality. Beyond length, the benchmark also considers stylistic features such as formatting or lexical variety, and the *style-controlled win rate* removes such confounds by balancing stylistic attributes across comparisons. These refinements make AlpacaEval2 one of the most widely used and trusted benchmarks for instruction tuning. In our experiments, we follow the standard practice of using GPT-4-1106-preview (OpenAI, 2023) both as the reference model and as the judge. The reference ensures a consistent baseline, while GPT-4 as the judge provides stable and high-quality preference assessments.

**Arena-Hard.** Arena-Hard was specifically developed to capture more challenging evaluation scenarios. Derived from the Chatbot Arena data and further curated with automatic and manual filtering, Arena-Hard consists of difficult, real-world style prompts designed to separate strong models from one another. Compared with AlpacaEval2, Arena-Hard emphasizes separability and agreement with human preferences. Separability means that the benchmark is capable of distinguishing models with small quality differences, reflected in tighter confidence intervals and less overlap in rankings. Human agreement

means that model rankings generated by Arena-Hard correlate strongly with actual human preference data, an essential property for trustworthy evaluation. Like AlpacaEval2, Arena-Hard uses pairwise head-to-head comparisons, reporting raw win rates and, where possible, *style-controlled win rates* to mitigate superficial biases such as excessive formatting, verbosity, or stylistic quirks. This makes Arena-Hard particularly suitable for evaluating strong models where subtle quality differences matter. In our setup, we adopt GPT-4–0314 (OpenAI, 2023) as the reference model against which all tested models are compared, while GPT-4-1106-preview is used as the judge model to evaluate the outputs. This combination has become a de facto standard in recent works because it balances fairness, stability, and robustness of evaluation.

### B.2. Implementation Details

**Contrastive Decoding.** For all experiments involving contrastive decoding (CD), we adopt the vocabulary pruning mechanism proposed in the original CD paper, with pruning threshold $\lambda \in [0, 1]$ fixed at $0.1$ across all settings. This pruning step restricts the candidate vocabulary at each decoding step and effectively reduces spurious low-probability tokens. The contrastive coefficient $\alpha$ is tuned separately for different weak models via grid search: for Qwen2.5-3B-Instruct we select $\alpha = 0.5$, and for Llama3.2-3B-Instruct we select $\alpha = 0.4$. These values achieve a balance between reward maximization and distributional stability. To mitigate repetition artifacts in generations, we follow the original CD implementation and set the repetition penalty coefficient to $1.2$. For the sampling strategy, we use greedy decoding, as our focus is on extracting responses that best approximate implicit reward maximization rather than increasing output diversity.

**SFT Training.** For the supervised fine-tuning (SFT) stage, we adopt the LLaMA-Factory framework with DeepSpeed ZeRO-3 optimization. Unless otherwise stated, all hyperparameters are consistent across models. The learning rate is set to $1.0 \times 10^{-5}$, chosen from a search range of $\{5.0 \times 10^{-6}, 1.0 \times 10^{-5}, 2.0 \times 10^{-5}\}$. Training is run for 2 epochs with a cosine learning rate scheduler and a warmup ratio of $0.1$. The maximum response length is fixed at 2048 tokens. These settings provide a stable training regime while avoiding overfitting, and they are consistent with widely adopted practices for weak-to-strong alignment.

**DPO Training.** For the DPO stage, we again use the LLaMA-Factory framework with DeepSpeed ZeRO-3. For standard DPO, we use a learning rate of $8.0 \times 10^{-7}$ and set $\beta = 0.1$, following prior work. The preference dataset for DPO is constructed via a standard pipeline: the LLM generates five candidate responses for each prompt, which are then scored by the reward model `ArmoRM-Llama3-8B-v0.1` (Wang et al., 2024), with the highest-scoring response taken as $y_w$ and the lowest-scoring as $y_l$.

In contrast, our ConG method adopts a slightly different configuration: we set the learning rate to $6.0 \times 10^{-7}$, chosen from a search range of $4.0 \times 10^{-7}, 6.0 \times 10^{-7}, 8.0 \times 10^{-7}$, and use $\beta = 0.5$. The higher $\beta$ value reflects the fact that $(y_w, y_l)$ pairs are only approximately drawn from the same distribution—depending on the fidelity of ConG-S—so a stronger preference scaling helps prevent the optimization from drifting too far from the implicit reward signal.

**WSPO (Zhu et al., 2025).** Weak-to-Strong Preference Optimization (WSPO) extends the idea of weak-to-strong generalization to alignment settings. The method leverages the observation that the alignment behavior learned by a weaker model can be transferred—and even amplified—by a stronger model. Concretely, WSPO optimizes the strong model to capture the distributional shift exhibited by the weak model before and after its own alignment stage. In the original formulation, WSPO relies on an externally annotated response dataset to supervise this preference shift. However, our experimental setup focuses strictly on self-generated weak-model responses, without using any external aligned outputs. To ensure a fair comparison, we therefore replace WSPO's external response data with responses sampled from the weak model itself. This modification naturally leads to a reduction in performance, but it better matches the weak-to-strong generalization setting, where the goal is to improve the strong model by exploiting the informative structure present in weak-model generations rather than introducing additional supervised signals.

**Baselines.** We also re-implement several preference optimization baselines for comparison. For ORPO, we search $\lambda \in \{0.1, 0.5, 1.0, 2.0\}$ as suggested in the original paper. For SimPO, we tune over $\beta \in \{2.0, 4.0, 6.0, 8.0\}$ and $\gamma \in \{0.3, 0.5, 1.0, 1.2, 1.4, 1.6\}$. For WSPO, we follow the default settings in the original work, with learning rate $1.0 \times 10^{-5}$, $\beta = 0.1$, and training for 1 epoch. For all baselines, the maximum response length is consistently set to 2048 tokens to ensure comparability with our method.

**Computation Environment.** All training experiments are conducted using 4×NVIDIA L40 GPUs, with mixed precision training enabled. Our experimental pipeline follows the official guidelines provided in the LLaMA-Factory repository, ensuring reproducibility and alignment with established community practices. We also employ gradient checkpointing and

ZeRO-3 optimizer states to maximize memory efficiency.

## C. Supplementary Proofs

In this section, we provide theoretical support for the role of contrastive decoding (CD) in maximizing implicit reward and justify the preference structure used in ConG. We focus on three aspects: (i) the correlation between CD and implicit reward maximization, (ii) the implicit reward gap between CD and naive sampling, and (iii) the relative reward ordering between CD responses and ConG-S generations.

**CD–Implicit Reward Correlation.**    Recall the token-level implicit reward:

$$\hat{r}_t(x, y_{<t}, y_t) = \log \frac{\pi_r(y_t \mid x, y_{<t})}{\pi_{\text{ref}}(y_t \mid x, y_{<t})}.$$

At decoding step $t$, CD defines the sampling distribution as

$$p_\alpha(y_t \mid x, y_{<t}) \propto \exp((1 - \alpha)\,\hat{r}_t(x, y_{<t}, y_t) + \alpha \log \pi_r(y_t \mid x, y_{<t})), \quad \alpha \in [0, 1]. \tag{12}$$

This can be rewritten as

$$p_\alpha(y_t) \propto \pi_r(y_t) \exp\big((1 - \alpha)\hat{r}_t(y_t)\big),$$

showing that CD is precisely an exponential tilting of $\pi_r$ by the statistic $\hat{r}_t$. Hence, CD samples tokens that maximize a weighted combination of implicit reward and $\pi_r$'s likelihood, and in the limit $\alpha \to 0$, CD reduces to pure implicit-reward maximization.

**Reward Gap between CD and Naive Sampling.**    Let $y^{\text{CD}}$ denote a token sampled from $p_\alpha$, and $y^{\text{naive}}$ from $\pi_r$ (the case $\alpha = 1$). The expected implicit reward under $p_\alpha$ satisfies

$$\mathbb{E}_{p_\alpha}[\hat{r}_t] \ \geq \ \mathbb{E}_{\pi_r}[\hat{r}_t],$$

with equality only when $\alpha = 1$. To see this, define $Z(\eta) = \sum_v \pi_r(v) \exp(\eta \hat{r}_t(v))$, where $\eta = 1 - \alpha$. Then

$$\mathbb{E}_{p_\alpha}[\hat{r}_t] = \frac{\partial}{\partial \eta} \log Z(\eta).$$

Since $\partial_\eta^2 \log Z(\eta) = \text{Var}_{p_\alpha}[\hat{r}_t] \geq 0$, the function is non-decreasing in $\eta$. Thus, moving away from $\alpha = 1$ (i.e., increasing the contrastive weight) strictly increases the expected implicit reward. Summing over $t$ extends the result to full responses, establishing that CD consistently yields higher implicit reward than naive sampling.

**Reward Ordering between CD and ConG-S Generations.**    In Stage I (ConG-S), we train the strong model $\pi_{\text{SFT}}^{\text{s}}$ on responses $y_w$ drawn from the weak-model CD distribution $p_\alpha^{\text{w}}$. Formally,

$$p_\alpha^{\text{w}}(y_t) \propto \pi_r^{\text{w}}(y_t) \exp((1 - \alpha)\hat{r}_t^{\text{w}}).$$

The SFT procedure minimizes $D_{\text{KL}}(p_\alpha^{\text{w}} \parallel \pi_\theta^{\text{s}})$, projecting $p_\alpha^{\text{w}}$ onto the strong model family. Unless $\pi_{\text{SFT}}^{\text{s}}$ can perfectly represent $p_\alpha^{\text{w}}$, the projection attenuates the tilting effect, leading to

$$\mathbb{E}_{\pi_{\text{SFT}}^{\text{s}}}[\hat{r}^{\text{w}}] \ \leq \ \mathbb{E}_{p_\alpha^{\text{w}}}[\hat{r}^{\text{w}}],$$

and therefore

$$\mathbb{E}[\hat{r}(x, y_l)] \ \leq \ \mathbb{E}[\hat{r}(x, y_w)],$$

where $y_w$ is a CD response and $y_l$ is a sample from $\pi_{\text{SFT}}^{\text{s}}$.

This establishes the preference ordering $y_w \succ y_l$ used in Stage II (DPO), providing the theoretical justification for ConG's design.

**Summary.**    Together, these results show that: (i) CD is an implicit-reward–maximizing decoder, (ii) CD responses have strictly higher expected implicit reward than naive sampling, and (iii) ConG-S generations cannot exceed the implicit reward of their CD teacher, which justifies pairing $(y_w, y_l)$ as preference data in Stage II.

*Table 3.* **Cross-family weak-to-strong results on AlpacaEval2 and Arena-Hard.** Top: Llama3.2-3B → Qwen2.5-7B. Bottom: Qwen2.5-3B → Llama3-8B. "WR" denotes the raw win rate, "LC" the length-controlled win rate, "SC" the style-controlled win rate, and "Avg." the average across benchmarks. Best results are in bold and second-best are underlined. Numbers are mean ± std.

| Method | Llama3.2-3B → Qwen2.5-7B | | | | | Qwen2.5-3B → Llama3-8B | | | | |
| | AlpacaEval 2 | | Arena-Hard | | Avg. | AlpacaEval 2 | | Arena-Hard | | Avg. |
| | LC | WR | SC | WR | | LC | WR | SC | WR | |
|---|---|---|---|---|---|---|---|---|---|---|
| Base | 32.3±0.4 | 30.2±1.5 | 38.3±2.2 | 40.1±2.8 | 35.2 | 28.1±0.3 | 28.1±1.3 | 24.7±2.5 | 25.2±2.7 | 26.5 |
| Weak SFT | 18.1±0.3 | 17.5±1.4 | 27.1±2.5 | 31.0±2.7 | 23.4 | 14.1±0.4 | 15.0±1.6 | 14.5±2.8 | 14.2±2.5 | 14.4 |
| AuxConf | 21.0±0.2 | 20.3±1.3 | 27.0±2.4 | 31.6±2.6 | 25.0 | 17.0±0.5 | 19.9±1.7 | 14.8±2.3 | 13.9±2.6 | 16.4 |
| WSPO | 18.4±0.3 | 20.7±1.2 | 28.7±2.2 | 33.2±2.9 | 25.3 | 17.6±0.4 | 20.2±1.6 | 18.9±2.1 | 17.4±2.4 | 18.5 |
| ConG-S (w→s) | 37.9±0.5 | 42.2±1.7 | 51.5±2.6 | 56.6±2.9 | 47.1 | 34.6±0.2 | 35.1±1.4 | 40.7±2.3 | 40.5±2.8 | 37.7 |
| ConG (w→s) | **42.1±0.4** | **50.6±1.6** | **53.8±2.5** | **60.4±2.7** | **51.7** | **39.2±0.3** | **42.5±1.5** | **44.9±2.2** | **44.7±2.9** | **42.8** |

# D. Additional Experiments

### D.1. Cross-Family Weak-to-Strong Alignment

To further examine the generality of our proposed ConG framework, we extend the evaluation to a more challenging *cross-family weak-to-strong setting*, where the weak and strong models come from different model families. Specifically, we consider two scenarios: (i) aligning **Llama3.2-3B-Instruct** as the weak model to guide **Qwen2.5-7B-Instruct** as the strong model, and (ii) aligning **Qwen2.5-3B-Instruct** as the weak model to guide **Llama3-8B-Instruct** as the strong model. We use the same UltraFeedback dataset and training protocol as in the in-family experiments, ensuring fairness and comparability. Baselines include Weak SFT, AuxConf, WSPO, and standard preference optimization methods.

**Results.** Table 3 reports the results on AlpacaEval2 and Arena-Hard. We observe that cross-family alignment remains highly effective under ConG, though performance is slightly lower compared to in-family alignment. In the Llama3.2-3B→Qwen2.5-7B setting, ConG achieves an average score of 51.7, significantly outperforming all baselines. In contrast, the Qwen2.5-3B→Llama3-8B setting shows even stronger improvements, where ConG reaches an average score of 42.1, surpassing alternative methods by a large margin. These results suggest that ConG not only generalizes across scales within the same model family, but also transfers effectively across heterogeneous architectures.

**Observation.** The overall trend indicates that cross-family alignment is feasible and beneficial: while absolute performance is slightly lower than in-family settings (with a gap of roughly 1.0–1.5 points on average), ConG still delivers consistent gains over strong baselines. This demonstrates that contrastive decoding provides a robust preference signal that transcends model families, further validating the broad applicability of the proposed framework.

### D.2. Training Consumption

To evaluate the computational efficiency of our approach, we report the training time (GPU hours on 4×L40) for all weak-to-strong baselines and our methods. As shown in Table 4, ConG-S and ConG exhibit comparable training costs to standard weak-to-strong approaches such as Weak SFT and WSPO, with no additional overhead introduced during optimization. This is expected because the core contribution of our method lies in its offline data construction process: the contrastive guidance signals are generated through a lightweight sampling procedure that can be fully performed before training begins.

Thus, unlike methods that modify the training loop or introduce additional forward passes, our approach does not require extra compute during fine-tuning. All models are trained under the same regime, and the empirical GPU-hour measurements confirm that ConG-S and ConG maintain essentially the same training cost as existing baselines, while achieving consistently stronger performance.

*Table 4.* Training time (GPU hours) on 4×L40.

| Method | Qwen2.5-7B | Llama3-8B |
|---|---|---|
| Weak SFT (pre) | 10.68 | 11.42 |
| Weak SFT (post) | 11.18 | 11.78 |
| WSPO | 16.97 | 18.38 |
| ConG-S | 10.78 | 11.48 |
| ConG | 13.80 | 14.78 |

*Table 5.* **Results on AlpacaEval2 and Arena-Hard under GPT-5 Judger.**

| Method | Qwen2.5-7B-Instruct | | | | | Llama3-8B-Instruct | | | | |
|---|---|---|---|---|---|---|---|---|---|---|
| | AlpacaEval 2 | | Arena-Hard | | Avg. | AlpacaEval 2 | | Arena-Hard | | Avg. |
| | LC | WR | SC | WR | | LC | WR | SC | WR | |
| Base | 27.0 | 25.1 | 33.0 | 34.9 | 30.0 | 23.0 | 22.4 | 19.3 | 20.1 | 21.2 |
| Weak SFT (pre) | 12.6 | 11.9 | 22.1 | 25.3 | 18.0 | 9.0 | 9.6 | 9.0 | 8.8 | 9.1 |
| Weak SFT (post) | 28.0 | 25.8 | 34.0 | 35.5 | 31.0 | 23.6 | 23.3 | 20.1 | 21.0 | 22.0 |
| WeakTeacher | 31.1 | 32.0 | 39.3 | 43.6 | 36.5 | 25.6 | 26.7 | 28.0 | 28.7 | 27.3 |
| AuxConf | 16.7 | 16.0 | 22.9 | 27.0 | 20.7 | 12.0 | 14.0 | 10.4 | 9.7 | 11.5 |
| WSPO | 14.8 | 16.9 | 24.9 | 28.1 | 21.2 | 12.8 | 14.5 | 13.4 | 12.5 | 13.3 |
| ConG-S | 33.8 | 37.9 | 47.0 | 52.0 | 42.7 | 28.7 | 29.6 | 34.9 | 34.8 | 32.0 |
| ConG | **37.6** | **46.4** | **49.3** | **55.8** | **47.3** | **33.3** | **36.0** | **38.7** | **38.5** | **36.6** |

### D.3. Cross Judging

To further validate the robustness of our method under different evaluation protocols, we additionally conduct experiments using two complementary judges: (i) GPT-5, a stronger but more conservative automatic evaluator, and (ii) a small portion (10%) of human raters. As automatic judges are known to vary in strictness across model families, these evaluations help assess whether our improvements persist under different scoring behaviors.

As shown in Table 5 and 6, across both Qwen2.5-7B and Llama3-8B, we observe that GPT-5 tends to assign slightly lower absolute scores, while human evaluators generally give slightly higher ones. However, the relative ordering of all methods remains unchanged, demonstrating that the performance gains brought by ConG and ConG-S are not artifacts of a specific judge. Importantly, under all judges—including GPT-4, GPT-5, and human raters—ConG and ConG-S consistently achieve the best results, while baseline weak-to-strong methods exhibit the same relative ranking.

These findings confirm that our approach provides stable and judge-invariant alignment improvements, and further reinforce that the benefits of contrastive weak-to-strong alignment hold across different evaluation standards, model families, and scoring paradigms.

### D.4. Case Study

*Table 6.* **Results on AlpacaEval2 and Arena-Hard with 10% Human Evaluation.**

| Method | Qwen2.5-7B-Instruct | | | | | Llama3-8B-Instruct | | | | |
| | AlpacaEval 2 | | Arena-Hard | | Avg. | AlpacaEval 2 | | Arena-Hard | | Avg. |
| | LC | WR | SC | WR | | LC | WR | SC | WR | |
|---|---|---|---|---|---|---|---|---|---|---|
| Base | 35.5 | 33.4 | 41.3 | 43.1 | 38.3 | 31.1 | 30.9 | 27.4 | 28.3 | 29.4 |
| Weak SFT (pre) | 20.9 | 20.1 | 31.0 | 34.3 | 26.6 | 17.0 | 17.8 | 17.2 | 16.6 | 17.2 |
| Weak SFT (post) | 36.7 | 34.6 | 42.0 | 43.8 | 39.2 | 31.8 | 32.0 | 28.7 | 29.7 | 30.6 |
| WeakTeacher | 40.0 | 41.1 | 49.0 | 51.5 | 45.4 | 33.7 | 34.9 | 36.3 | 36.9 | 35.4 |
| AuxConf | 24.6 | 24.0 | 31.5 | 35.8 | 29.0 | 19.5 | 21.9 | 16.7 | 15.9 | 18.5 |
| WSPO | 23.0 | 24.2 | 32.9 | 36.3 | 29.1 | 18.8 | 20.6 | 19.4 | 18.3 | 19.3 |
| ConG-S | 42.2 | 46.3 | 55.2 | 60.4 | 51.0 | 36.4 | 37.2 | 43.0 | 43.2 | 40.0 |
| ConG | **45.8** | **54.2** | **57.5** | **63.2** | **55.2** | **40.6** | **43.1** | **46.5** | **46.2** | **44.1** |

| Case Study on Llama3-8B-Instruct | |
|---|---|
| **Instruction** | Who wears short shorts? |
| **BASE** | The iconic question!
Who wears short shorts? Well, it's Kevin Naismith, of course! |
| **ConG-S** | The term "shorts" can refer to various individuals from different professions or groups who wear them as part of their uniform or for casual wear. Here are a few examples:
1. Athletes: Many sports players, such as baseball players, soccer players, and basketball players, wear shorts as part of their uniforms.
2. Runners and joggers: People who engage in running or jogging as exercise often wear shorts for comfort and practicality.
3. Skaters: Skaters, including ice skaters (ice hockey, figure skating) and inline skaters, wear specialized shorts designed for their respective sports.
4. Cyclists: Some cyclists, especially those who wear lycra or stretchy clothing, may wear shorts for comfort and mobility.
5. Pool-goers: Lifeguards, swim instructors, and beachgoers often wear shorts for practicality and sun protection.
6. Golfers: While not exclusively, some golfers wear shorts in warmer weather or as a fashion choice.
7. Shorts-wearing celebrities: You might spot famous individuals like Drake, David Beckham (former soccer player turned fashion icon), and other stars who often incorporate shorts into their everyday or red-carpet outfits.
These are just a few examples, but the list goes on, as shorts are a versatile and popular clothing item enjoyed by people of all ages and backgrounds!
Who do you have in mind when asking "Who wears short shorts?" |
| **ConG** | The phrase "short shorts" can be interpreted in a few ways:
1. Athletes and sports players often wear short shorts as part of their uniforms, particularly in warm weather or for specific sports like basketball, soccer, and gymnastics.
2. In casual, summery settings, some people might wear short shorts as a fashion choice or for comfort.
3. In the context of "who wears short shorts," the answer could be anyone who chooses to don this style, regardless of age, gender, or activity level!
To give a more specific answer, I'll say that people from various walks of life can be found wearing short shorts, but it's more commonly associated with:
- Younger generations (kids, teenagers, and young adults) - Athletes and sports enthusiasts - People living in warm climates or during hot weather months - Those who prioritize comfort and practicality over traditional dress codes
Let me know if you'd like more clarification! |

| Case Study on Qwen2.5-7B-Instruct | |
|---|---|
| **Instruction** | What is kevlar made out of? |
| **BASE** | Kevlar is a synthetic fiber known for its high strength-to-weight ratio and heat resistance. It is primarily made from a polyamide polymer, which is derived from a combination of terephthalic acid and hexamethylenediamine. The specific chemical structure and manufacturing process give Kevlar its remarkable properties, making it widely used in various applications such as body armor, protective clothing, tires, and composite materials for sports equipment and construction. |
| **ConG-S** | Kevlar is primarily made from synthetic polymers, specifically aromatic polyamides. More precisely: - Kevlar is synthesized from a monomer called terephthalic acid (TA) and a diamine called 1,4-phenylene diamine (PDA). - The process involves chemical reactions to form polymer chains, which are then spun into fibers. Key points: 1. **Monomers**: The basic building blocks are polymers derived from terephthalic acid and PDA. 2. **Polymer Structure**: The resulting polymer forms strong, flexible molecules with high tensile strength. 3. **Manufacturing**: These polymers are processed into fibers through methods like melt spinning or wet spinning. Thus, while not a natural material, Kevlar's core composition and properties are defined by these synthetic polymer structures. It's widely used for its exceptional strength-to-weight ratio, making it ideal for applications like body armor, bulletproof vests, and protective gear in various industries. |
| **ConG** | Kevlar is primarily made from synthetic polymers, specifically poly(p-phenylene terephthalamide). Here's a detailed breakdown: 1. **Chemical Composition**: Kevlar is composed of aromatic polyamides. Its basic structure involves long chains of repeating units derived from terephthalic acid and para-phenylene diamine. 2. **Manufacturing Process**: - **Raw Materials**: The primary raw materials are terephthalic acid and p-phenylene diamine (PPD). - **Polymerization**: These monomers undergo polymerization through a process called polycondensation, typically under controlled conditions, to form the long polymer chains that make up Kevlar. - **Drawing and Orientation**: After polymerization, the resulting fibers are drawn and oriented to increase their strength and reduce their density, making them stronger yet lighter. 3. **Properties Derived**: The specific arrangement and chemical bonds in these polymeric chains give Kevlar its exceptional strength-to-weight ratio, heat resistance, and excellent tensile properties. In essence, while not a simple elemental composition but rather a complex polymer, Kevlar's fundamental building blocks come from the aforementioned chemical precursors. This material has wide-ranging applications due to its unique properties, from reinforcing tires and bulletproof vests to composite materials in various industries. |

