# OpenReview forum: "Contrastive Weak-to-Strong Generalization"
_ICML.cc/2026/Conference — ICML 2026 regular_

### Official Review · Reviewer_Qovg · 2026-03-10

**Soundness:** 3
**Presentation:** 2
**Significance:** 3
**Originality:** 3
**Overall Recommendation:** 4
**Confidence:** 4

**Summary:**

In this paper, the authors propose a new framework called ConG (Contrastive Weak-to-Strong Generalization) to address the noise and bias issues in weak-to-strong alignment. The key idea is to establish a theoretical connection between the implicit reward in DPO and the mechanism of Contrastive Decoding (CD). Based on this connection, the authors utilize CD to generate higher-quality samples from the weak model to supervise the strong model.

**Compliance With Llm Reviewing Policy:**

Affirmed.

**Key Questions For Authors:**

Refer to W3 and W4

**Limitations:**

yes

**Strengths And Weaknesses:**

Strength:
1. The paper provides a solid motivation by revealing the structural equivalence between implicit rewards and Contrastive Decoding. This perspective is quite novel and provides a principled way to understand why CD can be used for preference alignment.
2. The authors conduct extensive experiments on Qwen2.5 and Llama3 families. Furthermore, the cross-family alignment results (e.g., Llama $\rightarrow$ Qwen) are very impressive. It demonstrates the generality of the proposed ConG framework across different architectures and tokenizers.
3. The contribution is also quite practical. Since the contrastive decoding process is performed offline during the data construction stage, it does not introduce additional computational overhead during the actual training phase.

Weakness:
1. I noticed a very unprofessional clerical error in the manuscript. The running header from page 2 to page 20 reads "Training-free Streaming Safeguard with Sparse Autoencoders," which is totally irrelevant to the title of this paper.
2. The evaluation is mainly based on AlpacaEval2 and Arena-Hard using GPT-4 as the judge. It is well known that LLM judges often have a "verbosity bias" or "style preference." Since Contrastive Decoding changes the token distribution to be more "sharp," it is possible that the gain comes from satisfying the judge's stylistic bias rather than real capability improvement. I suggest the authors provide more results on objective, ground-truth benchmarks or human evaluations.
3. According to Figure 3, the performance of ConG is quite sensitive to the contrastive coefficient $\alpha$. When $\alpha$ exceeds 0.5, the performance drops significantly. This indicates that the optimal $\alpha$ may vary across different model pairs, which might limit the robustness of the method when applied to new, unknown models. The authors should provide more discussion on how to choose $\alpha$ in a more principled way.
4. The paper lacks a deep analysis of the "lower bound" of the weak model. If the weak model is extremely poor or has a very large capability gap with the strong model, will the contrastive signal still be informative? Adding some "failure case" analysis would make the paper more complete.

---

> ### Author Rebuttal · Authors · 2026-03-31
>
> Dear Reviewer Qovg:
>
> Thank you for your positive feedback and valuable suggestions! We sincerely appreciate the time and effort you have dedicated to reviewing our work. Below, we meticulously provide responses to each of your comments and outline the modifications based on your suggestions.
>
> ## *W1: The running header from page 2 to page 20 is totally irrelevant to the title of this paper.*
>
> Thank you for pointing this out. This was indeed our oversight. In the revised version, we have corrected the running header to match the title of this paper and carefully proofread the manuscript again. During this revision, we also fixed several imprecise or overly strong expressions, including:
>
> 1. **"preserve the full signal of target preferences"**
>    Revised to: **"preserve target-preference signals to some extent"**.
>
> 2. **"approximately maximize implicit reward"**
>    Revised to: CD **biases decoding toward higher implicit-reward responses**.
>
> 3. **Repeated use of "equivalence"**
>    Revised to: **"correlation"**.
>
> Hope our response could address your concerns!
>
> ## *W2: I suggest the authors provide more results on objective, ground-truth benchmarks or human evaluations.*
>
> Thank you for this important suggestion. We agree that relying only on LLM-based judges is insufficient, and we have already included additional evidence from both **objective, ground-truth benchmarks** and **cross-judge / human evaluation**.
>
> 1. **Objective, ground-truth benchmarks.**
>    In original `Appendix D.2` (Lines 813--857)**, we further evaluate alignment on **math and coding tasks**, which provide objective, ground-truth outcomes rather than judge-based preferences.
>
> 2. **Cross-judge and human evaluation.**
>    In original `Appendix D.5` (Lines 904--917), we also include cross-judging results using **GPT-5** as an additional judge, together with **10\% subset human evaluation**, to verify that the improvement is robust beyond GPT-4-based evaluation.
>
> Hope our response could address your concerns!
>
>
> ## *W3: The authors should provide more discussion on how to choose $\alpha$ in a more principled way.*
>
> Thank you for this helpful suggestion. We agree that the choice of $\alpha$ should be discussed more clearly. In fact, this is already analyzed in the original manuscript from two perspectives:
>
> 1. `Section 3.3` studies how $\alpha$ affects implicit reward, explicit reward, and potential length bias during CD generation.
> 2. `Section 5.4` further examines how different $\alpha$ values affect the final performance of ConG.
>
> Across model pairs, our results show a consistent pattern: **$\alpha \in [0,0.5]$ generally works best**, while performance drops noticeably when $\alpha > 0.5$. Intuitively, smaller $\alpha$ preserves stronger contrastive preference signal, whereas larger $\alpha$ makes decoding closer to standard decoding.
>
> In practice, we choose $\alpha$ by **sampling a small held-out subset on the weak side** and inspecting the distribution of explicit reward and implicit reward (as in Figure 2), then selecting the $\alpha$ that gives the **highest overall explicit reward** while maintaining stable behavior.
>
> We will revise the manuscript to make this practical selection rule more explicit.
>
> ## *W4: The paper lacks a deep analysis of the "lower bound" of the weak model.*
>
> Thank you for this valuable comment. We agree that the weak model’s lower bound is important to understand. In fact, the original `Section 5.3` and `Appendix D.4` already analyze how performance changes with different weak–strong capability gaps. In particular, when the weak model is very limited, its ability to enhance a much stronger model is indeed constrained.
>
> To make this point more explicit, we additionally summarize the average performance of different base models and their CD-decoded outputs across model sizes:
>
> |Family|Model|Base Avg.|CD Avg.|
> |:-:|:-:|:-:|:-:|
> |Qwen2.5|1.5B|18.9|33.4|
> ||3B|22.8|43.5|
> ||7B|35.2|58.8|
> |Llama3|1B|15.6|24.9|
> ||3B|21.8|37.8|
> ||8B|26.5|50.9||||||
>
> These results show that CD consistently improves the weak model itself, but when the gap is too large, the CD signal still has a practical limit. For example, Qwen2.5-1.5B + CD still does not clearly surpass the Qwen2.5-7B base model, which helps explain why directly using a 1.5B weak model to enhance a 7B strong model brings limited gains.
>
> Based on this, our recommendation is a **step-wise enhancement strategy**, e.g., **1.5B $\rightarrow$ 3B, 3B $\rightarrow$ 7B, and 7B $\rightarrow$ 14B**, rather than directly enhancing a very strong model from an extremely weak one. We believe this is a more realistic and principled path toward weak-to-strong generalization.
>
> Hope our response could address your concerns!
>
> ------------
>
> *Once again, we are deeply appreciative of the time and expertise you have shared with us, and we are more than happy to add clarifications to address any additional recommendations and reviews from you!*
>
> Best,
> Authors of Submission 24640

---

> > ### Author Rebuttal · Reviewer_Qovg · 2026-04-05
> >
> > I will remain my score

---

> > > ### Author Response · Authors · 2026-04-08
> > >
> > > Dear Reviewer Qovg,
> > >
> > > Thank you very much for your thoughtful engagement and for maintaining your positive score. We sincerely appreciate your time, support, and constructive feedback throughout the discussion.
> > >
> > > We are glad that our responses helped clarify the paper. To briefly restate, we believe the core contribution of our work is to **identify a new transferable weak supervision signal for weak-to-strong generalization—namely, the contrastive signal between pre- and post-improved weak models—and to connect this signal to implicit reward both theoretically and empirically**.
> > >
> > > We are also grateful for your comments because they helped us sharpen the presentation and positioning of the paper. More broadly, we hope this work can contribute to the W2SG community by opening up new directions for extracting scalable supervision from weak models and transferring it more effectively to stronger ones.
> > >
> > > Thank you again for your support. If you have any further questions or concerns, we would be very happy to address them.
> > >
> > > Best,
> > >
> > > Authors of Submission 24640

---

### Official Review · Reviewer_QHdF · 2026-03-10

**Soundness:** 2
**Presentation:** 3
**Significance:** 3
**Originality:** 2
**Overall Recommendation:** 3
**Confidence:** 4

**Summary:**

This paper investigates the quality of supervision signals in weak-to-strong generalization. The authors propose ConG, which uses contrastive decoding between pre-aligned weak models and post-aligned weak models to generate higher-quality training samples, followed by a two-stage training of the strong model. The first stage, ConG-S, performs SFT using the contrastive decoding-generated samples; the second stage, ConG, constructs preference pairs and applies DPO. The core idea of the paper is that the implicit reward's log-likelihood ratio structure aligns with the scoring form of contrastive decoding, meaning contrastive decoding can be understood as a mechanism that helps produce samples with high implicit reward, thereby improving weak-to-strong generalization. Experiments are conducted on two model families, Qwen2.5 and Llama3, with results mainly reported on AlpacaEval2, Arena-Hard, several downstream tasks, and math/code tasks. The authors conclude that ConG achieves better instruction-following performance compared to several weak-to-strong baselines while largely preserving general capabilities.

**Compliance With Llm Reviewing Policy:**

Affirmed.

**Key Questions For Authors:**

See Weaknesses

**Limitations:**

Yes

**Strengths And Weaknesses:**

### Strengths

- The core idea is simple and implementable: use contrastive decoding between pre- and post-alignment weak models to “denoise” supervision, then use the resulting samples in a two-stage SFT plus DPO pipeline.

- In addition to the main experimental results, the paper also provides detailed reports on the influence of the contrastive weight parameter $\alpha$, the preservation of downstream general capabilities, performance on mathematics and coding tasks, effects under cross-model-family settings, analysis of training costs, and experimental outcomes with larger model scales. This demonstrates that the conclusions of the paper are not solely reliant on a single main table but are supported by a comprehensive, multi-dimensional analysis.

- The issue investigated in this paper is significant. Weak-to-strong generalization and scalable alignment are indeed important directions of current research, especially in scenarios where high-quality human feedback is expensive or even unavailable.

### Weaknesses

- The paper employs numerous strong expressions, such as "samples obtained via contrastive decoding preserve the full signal of target preferences and approximately maximize implicit reward," and assertions in the conclusion like "This equivalence allows us…" The issue lies in the fact that what the paper actually supports appears to be more consistent with the idea that the scoring form of CD aligns structurally with implicit reward, thereby inclining CD to prefer candidates with higher implicit reward, rather than having fully demonstrated the stronger proposition that "CD-generated samples are equivalent to maximizing implicit reward."

- In Appendix D.3 (Table 5), the paper attempts to demonstrate that ConG does not increase "training consumption." However, this approach appears to sidestep the primary issue. The main cost of ConG lies in the data generation phase: it requires loading both the pre-aligned and post-aligned weak models into memory simultaneously and performing autoregressive contrastive decoding. This process incurs significantly higher latency and memory overhead compared to standard sampling or Best-of-N methods. The paper completely omits the quantification of GPU hours and throughput during this stage, which is an unacceptable omission when evaluating the engineering scalability of the method in real-world scenarios.

- The method constructs DPO preference pairs by assuming that the contrastive decoding-generated response \(y_w\) should be preferred over the supervised fine-tuning-generated response \(y_l\), and this ordering is explicitly encoded in the algorithm. However, the paper does not directly report the empirical validity of this assumption under independent evaluation or human annotation, nor does it examine the stability of this ordering across different tasks, \(\alpha\) values, and model configurations. Additionally, since \(y_w\) and \(y_l\) originate from distinct generation processes, this setup may introduce covariate shift concerns that are not analyzed.

- This work resembles a valuable form of combinatorial innovation and theoretical reinterpretation, rather than a breakthrough in entirely new, strong theories or novel mechanisms. Contrastive decoding itself is not a new technique; implicit reward has also been studied previously; and weak-to-strong alignment likewise has a considerable body of prior work. The core contribution of this paper lies in effectively integrating these existing components and constructing a framework that proves effective in practice.

- Multiple pages display an unrelated running header (“Training-free Streaming Safeguard with Sparse Autoencoders”), which suggests that the manuscript was not carefully proofread.

---

> ### Author Rebuttal · Authors · 2026-03-31
>
> Dear Reviewer QHdF,
>
> Thank you for your careful review and constructive suggestions. We address each concern below and summarize the corresponding revisions.
>
> ## *W1: The paper employs numerous strong expressions.*
>
> Thank you for pointing this out. We agree that some claims were overstated, and we revised the manuscript accordingly:
>
> 1. **"preserve the full signal of target preferences"**
>    Revised to: **"preserve target-preference signals to some extent"**.
>
> 2. **"approximately maximize implicit reward"**
>    Revised to: CD **biases decoding toward higher implicit-reward responses**.
>
> 3. **Repeated use of "equivalence"**
>    Revised to:  **"correlation"**.
>
> ## *W2: This CD process incurs significantly higher latency and memory overhead compared to standard sampling or Best-of-N methods.*
>
> We agree that the main extra cost of ConG lies in the **offline CD data-generation stage**, which was not clearly quantified in the original Appendix D.3. In response:
>
> 1. **Efficiency.** CD is slower than standard sampling because it contrasts two model  during decoding. In our setup, generating **600K CD samples takes 9.6 hours on 8×A800 GPUs**, while standard sampling takes **2.6 hours** under the same token budget. BoN-4 requires about **10.5 hours** before reranking. Thus, CD is slower than standard sampling, but comparable to BoN in generation cost. More importantly, this is a **one-time offline cost**: once generated, the same CD data can be reused to train multiple downstream strong models.
>
> 2. **Effectiveness.** We additionally compare ConG against **BoN with explicit reward (BoN+ER)**, **BoN with weak-model implicit reward (BoN+IR)**, and **weak-model BoN data used directly for strong-model training (Weak-BoN)**:
>
> |Model|Method|AE-LC|AE-WR|AH-SC|AH-WR|
> |:-:|:-:|:-:|:-:|:-:|:-:|
> |Qwen2.5-7B|Base|32.3|30.2|38.3|40.1|
> ||Weak-BoN|18.5|18.1|28.1|32.2|
> ||BoN+ER|36.5|38.9|46.8|52.3|
> ||BoN+IR|35.2|36.8|44.7|49.6|
> ||ConG-S|38.7|43.1|52.4|57.8|
> ||ConG|43.0|51.9|54.5|61.2|
> |Llama3-8B|Base|28.1|28.1|24.7|25.2|
> ||Weak-BoN|14.4|15.1|14.8|14.3|
> ||BoN+ER|31.6|32.8|34.8|35.5|
> ||BoN+IR|30.2|31.3|32.1|32.8|
> ||ConG-S|33.7|34.3|39.6|39.2|
> ||ConG|38.3|41.2|43.6|43.3|
> |||||||
>
> ConG consistently outperforms all BoN-methods, suggesting that its gains come from the preference-relevant signal introduced by CD rather than merely extra sampling budget.
>
> We have added these cost statistics and BoN comparisons to the manuscript.
>
> ## *W3: The paper does not directly validate whether the assumed ordering $y_w \succ y_l$ is empirically valid and stable across different $\alpha$ values.*
>
> We agree that the assumed ordering $y_w \succ y_l$ should be validated more directly. In fact, original `Appendix C` (**Lines 751–765**) already provides a theoretical argument showing that, **in expectation, the implicit reward of $y_w$ is higher than that of $y_l$**. To complement this, we now report the proportion of cases where $y_w$ is preferred over $y_l$ under explicit reward (ER), implicit reward (IR), and human evaluation on a subset, across different $\alpha$ values:
>
> |$\alpha$|ER|IR|Human|
> |:-:|:-:|:-:|:-:|
> |0.0|0.76|0.93|0.82|
> |0.2|0.75|0.91|0.80|
> |0.4|0.74|0.88|0.79|
> |0.6|0.66|0.84|0.68|
> |0.8|0.60|0.72|0.58|
> |1.0|0.55|0.53|0.51|
> |||||
>
> These results are consistent with our theory: **IR most directly matches the predicted ordering**, and **ER/human evaluation follow a similar trend**, with smaller $\alpha$ yielding a clearer advantage for $y_w$. This directly supports our pair construction. Regarding covariate shift, ConG also mitigates distribution mismatch by first using ConG-S to SFT-train the strong model on CD responses before DPO.
>
> ## *W4: The work appears more as a combinatorial integration than a fundamentally new mechanism.*
>
> Thank you for this comment. Our contribution is mainly theoretical analysis and framework-level. Specifically:
> 1. We are the **first to identify and formalize the connection between implicit reward and contrastive decoding**, offering a new perspective on why CD improves generation quality.
> 2. We support this connection with both **theoretical analysis and empirical evidence**, showing that CD amplifies preference-relevant signals and can serve as a practical mechanism for weak-to-strong transfer.
> 3. We **broaden weak-to-strong generalization** by showing that, without extra reward models, CD-generated data can support both weak-to-strong alignment and self-alignment.
>
>
> ## *W5: Multiple pages display an unrelated running header, suggesting insufficient proofreading.*
>
> Thank you for pointing this out. This was our oversight during manuscript preparation. We have corrected the unrelated running header and carefully proofread the revised manuscript.
>
> ------------
>
> *Once again, we are deeply appreciative of the time and expertise you have shared with us, and we are more than happy to add clarifications to address any additional recommendations and reviews from you!*
>
> Best,
> Authors of Submission 24640

---

> > ### Author Rebuttal · Reviewer_QHdF · 2026-04-05
> >
> > Thanks to the authors for providing answers to my questions. I still have concerns that this work appears more as a combinatorial integration than a fundamentally new mechanism. I will maintain my score due to the core contribution.

---

> > > ### Author Response · Authors · 2026-04-05
> > >
> > > Dear Reviewer QHdF,
> > >
> > > Thank you for your follow-up and for continuing to recognize the paper’s core contribution. We sincerely appreciate that support. We understand that your remaining concern is whether ConG is more of a combinatorial integration than a fundamentally new mechanism. We would like to clarify this from three perspectives.
> > >
> > > ## 1. In W2SG, the key question is how to extract better weak supervision
> > >
> > > We agree that ConG follows the common W2SG paradigm of **weak-model data generation + strong-model training**. However, this is also true for most representative W2SG methods:
> > >
> > > ```text
> > > Weak-SFT [1]    : weak-model sampling                 + SFT
> > > WeakTeacher [2] : weak-model implicit-reward ranking  + DPO
> > > W2SG-Mo [3]     : weak-model sampling/filtering       + SFT
> > > MACPO [4]       : multi-agent weak-data synthesis     + SFT
> > > ConG (ours)     : weak-model contrastive decoding     + SFT / DPO
> > > ```
> > >
> > > Thus, in W2SG, the main contribution is usually **not** inventing a brand-new optimizer, but finding a **better way to mine useful supervision from weak models**. The original W2SG line itself is valuable for this reason: it studies how weak supervision can be turned into effective strong-model improvement. We believe ConG should be evaluated in this same context. Our key contribution is to show that **contrastive weak-pair signal** is a more informative supervision source than direct weak outputs alone.
> > >
> > > ## 2. Replacing weak-data generation with CD brings clear and consistent gains
> > >
> > > To further support this point, we additionally compare with newer related baselines, including **W2SG-Mo** and **MACPO**:
> > >
> > > |Model|Method|Align.Avg.|Math.Avg.|Code.Avg.|
> > > |:-:|:-:|:-:|:-:|:-:|
> > > |Qwen2.5-7B|Base|35.2|74.0|77.3|
> > > ||Weak-SFT|35.9|74.6|77.8|
> > > ||WeakTeacher|43.5|76.4|78.9|
> > > ||W2SG-Mo|45.1|75.8|78.5|
> > > ||MACPO|46.3|77.0|79.1|
> > > ||ConG-S|48.0|77.7|80.1|
> > > ||ConG|52.7|79.2|81.0|
> > > |Llama3-8B|Base|26.5|59.7|65.1|
> > > ||Weak-SFT|27.2|60.5|65.5|
> > > ||WeakTeacher|32.6|61.9|66.7|
> > > ||W2SG-Mo|33.8|61.1|66.2|
> > > ||MACPO|35.0|62.5|66.9|
> > > ||ConG-S|36.7|63.4|68.2|
> > > ||ConG|41.6|64.5|69.0|
> > >
> > > These results show that, although ConG is simple in form, **replacing the first-stage weak-data generation with contrastive decoding already yields substantial and consistent gains** over both earlier and newer W2SG baselines. This is an important point: if the method were merely a superficial recombination of familiar components, then baselines built from similar ingredients should already close the gap. Empirically, they do not.
> > >
> > > So we agree that ConG is simple, but we believe the evidence shows it is **not merely combinatorial**. The simplicity is actually part of the contribution: once the right weak supervision signal is identified, strong gains can be obtained without adding a complex new training stack.
> > >
> > > ## 3. The deeper contribution is theoretical and conceptual
> > >
> > > More importantly, our contribution is not only a practical combination:
> > >
> > > - **First**, to our knowledge, this is the first work to explicitly connect **contrastive decoding** with **implicit reward** in the W2SG setting. This gives a principled explanation for why CD can expose preference-relevant signal, rather than behaving only as a decoding trick.
> > > - **Second**, to our knowledge, this is also the first work to propose transferring the **contrast between weak-model states**, rather than only raw weak-model outputs. This is the central conceptual step of the paper.
> > > - **Third**, this naturally leads to a new W2SG paradigm: use the **improvement-induced weak contrastive signal** to supervise stronger models.
> > >
> > > In this sense, the pipeline is the realization of the idea, not the idea itself. The core novelty is discovering that the **contrast between pre- and post-improved weak models is itself a transferable supervision signal**, and then showing both theoretically and empirically that this signal can systematically improve stronger models.
> > >
> > > ---
> > >
> > > In summary, we agree that ConG is implemented with familiar components. However, in the context of W2SG, we respectfully believe the paper contributes:
> > >
> > > - a **new theoretical connection** between CD and implicit reward,
> > > - a **new supervision perspective** for weak-to-strong transfer,
> > > - and a **simple but consistently stronger framework** built on that principle.
> > >
> > > For these reasons, we believe ConG is more than a procedural integration and represents a meaningful conceptual contribution to the W2SG literature.
> > >
> > > Thank you again for your thoughtful engagement and continued support.
> > >
> > > Best,
> > >
> > > Authors of Submission 24640
> > >
> > > [1] *Weak-to-Strong Generalization: Eliciting Strong Capabilities With Weak Supervision.* ICML 2024.
> > > [2] *Your Weak LLM is Secretly a Strong Teacher for Alignment.* ICLR 2025.
> > > [3] *How to Mitigate Overfitting in Weak-to-Strong Generalization?* ACL 2025.
> > > [4] *MACPO: Weak-to-Strong Alignment via Multi-Agent Contrastive Preference Optimization.* ICLR 2025.

---

### Official Review · Reviewer_Yg5S · 2026-03-12

**Soundness:** 2
**Presentation:** 3
**Significance:** 2
**Originality:** 2
**Overall Recommendation:** 4
**Confidence:** 2

**Summary:**

This paper proposes Contrastive Weak-to-Strong Generalization (ConG), a simple two-stage framework that uses contrastive decoding (CD) between pre- and post-alignment weak models to generate higher-quality supervision for training a stronger model. The authors establish a mathematical link between token-level implicit rewards (log-likelihood ratios) and contrastive decoding, argue that CD approximately maximizes implicit reward, and then exploit CD-generated responses to (i) initialize the strong model with SFT (ConG-S) and (ii) further refine it with DPO using CD responses as “chosen” references. Experiments on Qwen2.5 and Llama3 families across AlpacaEval2 and Arena-Hard show consistent gains over several weak-to-strong and preference-optimization baselines, with additional analyses on the effect of the contrastive coefficient, model capability gaps, cross-family transfer, and downstream utility preservation.

**Compliance With Llm Reviewing Policy:**

Affirmed.

**Final Justification:**

Most of issues have been resolved.

**Key Questions For Authors:**

1. How would ConG fare against a simple best-of-N baseline using (a) an external reward model, and (b) implicit-reward reranking (log π_post − log π_pre) of weak-model samples? This would help isolate the value of CD versus simpler selection strategies.

2. Can you provide compute/latency statistics for CD pre-generation and the overall ConG pipeline relative to standard weak-to-strong baselines? How does α affect cost and acceptance rates under vocabulary pruning?

3. How robust are the gains to the choice of judge model? Do rankings persist under a different LLM judge or a small-scale human evaluation? Any inter-judge agreement statistics?

**Limitations:**

yes

**Strengths And Weaknesses:**

Strengths:
1. The two-stage pipeline (ConG-S + ConG) is simple, modular, and practical: pre-generate CD responses once from weak models, SFT the strong model on them, then apply DPO using a paired sample scheme that leverages the implicit-reward advantage of CD generations.

2. The core idea and pipeline are easy to grasp, with intuitive figures and a reasonable derivation detailing how CD steers toward higher implicit reward.

3. Weak-to-strong generalization is an important and timely setting; providing a robust way to denoise weak-model supervision without external reward models addresses a key bottleneck.

Weaknesses:
1. It feels like an indirect or roundabout way of training the weak model. The premise of Weak-to-Strong (W2SG) should be the assumption that no higher-quality data source is available, and the goal is to extract valuable signals from the data generated by the weak model itself.

2. The theoretical justification for using CD responses as “chosen” in DPO relies on expected implicit-reward ordering and KL projection arguments; this is plausible but high-level, and does not address potential distributional mismatch or reinforcement of weak-model biases.

3. The paper does not compare to simple but competitive denoising strategies for the weak teacher such as best-of-N sampling with an external reward model, best-of-N with implicit reward reranking (log-ratio-based scoring), or rejection sampling with an off-the-shelf reward model. These are natural and important baselines to judge whether CD is the key factor.

4. Heavy reliance on GPT-4-based judging without a human evaluation subset or alternative judges raises concerns about evaluator bias and robustness of the reported gains.

---

> ### Author Rebuttal · Authors · 2026-03-31
>
> Dear Reviewer Yg5S:
>
> Thank you for your careful review and constructive suggestions. We address each concern below and summarize the corresponding revisions.
>
> ------------
>
> ## *W1: The premise of W2SG should be the assumption that no higher-quality data source is available.*
>
> Thank you for raising this point. In response:
>
> - **ConG only needs pre/post-aligned weak models.** ConG uses two weak models, and performs CD from their log-ratio. It does not require access to the higher-quality data.
>
> - **We use external data to train the post-aligned weak model ourselves only for a controlled and fair comparison with prior W2SG baselines.** This is an evaluation choice, not a requirement of ConG. **This is also consistent with prior W2SG baselines such as WeakTeacher [1] and WSPO [2].**
>
> Hope our response could address your concerns!
>
> [1] Your Weak LLM is Secretly a Strong Teacher for Alignment. ICLR 2025.
> [2] Weak-to-Strong Preference Optimization: Stealing Reward from Weak Aligned Model. ICLR 2025.
>
> ## *W2. Using CD responses as ‘chosen’ in DPO does not address distribution mismatch or weak-model bias.*
>
> Thank you for raising this concern. We provide our response from the following two perspectives:
> - **ConG is explicitly designed to mitigate distribution mismatch.** We do not apply DPO on the base strong model directly. Instead, **Stage 1 (ConG-S) first performs SFT on CD responses, moving the strong model toward the CD-induced distribution;** Stage 2 then applies DPO on top of this initialization. Thus, the strong-model rollouts used in DPO are already much closer to the CD responses.
>
> - **CD responses are not naive weak samples, and our results do not support mere bias amplification.** ConG does not use raw weak-model outputs, but CD responses generated from the contrastive signal between pre- and post-alignment weak models, which better isolates alignment-induced preference signal rather than simply copying weak outputs.
>
> Hope our response could address your concerns!
>
> ## *W3&Q1: The paper does not compare to simple but competitive denoising strategies for the weak teacher.*
>
> Thank you for this helpful suggestion. Following your suggestion:
>
> - **We have added the BoN-based baselines** :
>   (1) BoN (Best-of-4) with explicit reward model (BoN+ER),
>   (2) BoN (Best-of-4) with weak-model implicit-reward (BoN+IR).
>   The results are shown in the table below:
>
> |Model|Method|AE-LC|AE-WR|AH-SC|AH-WR|
> |:-:|:-:|:-:|:-:|:-:|:-:|
> |Qwen2.5-7B|Base|32.3|30.2|38.3|40.1|
> ||BoN+ER|36.5|38.9|46.8|52.3|
> ||BoN+IR|35.2|36.8|44.7|49.6|
> ||ConG-S|38.7|43.1|52.4|57.8|
> ||ConG|43.0|51.9|54.5|61.2|
> |Llama3-8B|Base|28.1|28.1|24.7|25.2|
> ||BoN+ER|31.6|32.8|34.8|35.5|
> ||BoN+IR|30.2|31.3|32.1|32.8|
> ||ConG-S|33.7|34.3|39.6|39.2|
> ||ConG|38.3|41.2|43.6|43.3|
> |||||||
>
> - **The new results show that ConG still achieves the most robust performance against these strong denoising baselines.** This suggests that the gain of ConG is from more effectively refining weak-model supervision through contrastive decoding.
>
> Hope our additional experiments could resolve your concern!
>
> ## *W4&Q3: Heavy reliance on GPT-4-based judging without a human evaluation subset or alternative judges.*
>
> Thank you for this important concern. We fully agree that cross-judging with alternative judges and human evaluation is necessary for robust evaluation.
>
> In fact, this is already included in original `Appendix D.5` (**Lines 904-917**): we report results with GPT-5 and a 10% human evaluation subset. The results show that the overall ranking remains consistent, with **ConG/ConG-S staying best overall**.
>
> ## *Q2: Can you provide compute/latency statistics for CD pre-generation and the overall ConG pipeline? How does $\alpha$ affect cost and acceptance rates under vocabulary pruning?*
>
> Thank you for this helpful question. In response:
>
> - CD is slower than standard sampling because it contrasts two model  during decoding. In our setup, generating **600K CD samples takes 9.6 hours on 8×A800 GPUs**, while standard sampling takes **2.6 hours** under the same token budget. BoN-4 requires about **10.5 hours** before reranking. Thus, CD is slower than standard sampling, but comparable to BoN in generation cost. More importantly, this is a **one-time offline cost**: once generated, the same CD data can be reused to train multiple downstream strong models.
>
> - **$\alpha$ mainly affects decoding selectivity rather than overall pipeline cost.** Under vocabulary pruning, different $\alpha$ values mainly change the **kept-token ratio**, which we summarize in the table below：
>
> |Alpha|KeptTokenRatio(%)|
> |:-:|:-:|
> |0.0|86.1|
> |0.2|90.7|
> |0.4|93.1|
> |0.6|97.4|
> |0.8|99.5|
> |1.0|100.0|
> |||
>
> Hope our response could address your concerns!
>
> ------------
>
> *Once again, we are deeply appreciative of the time and expertise you have shared with us, and we are more than happy to add clarifications to address any additional recommendations and reviews from you!*
>
> Best,
>
> Authors of Submission 24640

---

> > ### Author Rebuttal · Reviewer_Yg5S · 2026-04-03
> >
> > The author added BoN results. It demonstrates that are indeed non-increment work. However, I'm still not entirely convinced that this is a fundamentally new idea, rather than just a clever decoding and training pipeline. I will lower my current confidence and provide as much positive support as possible.

---

> > > ### Author Response · Authors · 2026-04-04
> > >
> > > Dear Reviewer Yg5S,
> > >
> > > Thank you for your follow-up and encouraging note. We are glad that the added BoN comparisons helped clarify that the paper is not merely incremental, and we sincerely appreciate your willingness to provide positive support. At the same time, we understand that one concern remains: whether ConG should be viewed as a fundamentally new idea, rather than only a clever decoding-and-training pipeline. We would like to clarify this from three perspectives.
> > >
> > > ## 1. In W2SG, the key question is how to extract better weak supervision
> > >
> > > We agree that ConG follows the common W2SG pattern of **weak-model data generation + strong-model training**. However, this is also true for most representative W2SG methods:
> > > ```text
> > > Weak-SFT [1]    : weak-model sampling                 + SFT
> > > WeakTeacher [2] : weak-model implicit-reward ranking  + DPO
> > > W2SG-Mo [3]     : weak-model sampling/filtering       + SFT
> > > MACPO [4]       : multi-agent weak-data synthesis     + SFT
> > > ConG (ours)     : weak-model contrastive decoding     + SFT / DPO
> > > ```
> > > Thus, in W2SG, the main contribution is usually **not** inventing a brand-new optimizer, but discovering a **better way to mine useful supervision from weak models**. We believe ConG should be evaluated in this same context. Our key contribution is to show that the **contrastive weak-pair signal** is substantially more informative than direct weak outputs alone.
> > >
> > > ## 2. Replacing weak-data generation with CD brings clear gains
> > >
> > > To further support this point, we additionally compare against newer related baselines, including **W2SG-Mo** and **MACPO**:
> > >
> > > |Model|Method|Align.Avg.|Math.Avg.|Code.Avg.|
> > > |:-:|:-:|:-:|:-:|:-:|
> > > |Qwen2.5-7B|Base|35.2|74.0|77.3|
> > > ||Weak-SFT|35.9|74.6|77.8|
> > > ||WeakTeacher|43.5|76.4|78.9|
> > > ||W2SG-Mo|45.1|75.8|78.5|
> > > ||MACPO|46.3|77.0|79.1|
> > > ||ConG-S|48.0|77.7|80.1|
> > > ||ConG|52.7|79.2|81.0|
> > > |Llama3-8B|Base|26.5|59.7|65.1|
> > > ||Weak-SFT|27.2|60.5|65.5|
> > > ||WeakTeacher|32.6|61.9|66.7|
> > > ||W2SG-Mo|33.8|61.1|66.2|
> > > ||MACPO|35.0|62.5|66.9|
> > > ||ConG-S|36.7|63.4|68.2|
> > > ||ConG|41.6|64.5|69.0|
> > >
> > > Although ConG is simple in form, **replacing the first-stage weak-data generation with contrastive decoding already yields large and consistent gains** over both earlier and newer W2SG baselines. If the method were only a superficial recombination of familiar components, then baselines built from similar ingredients should largely close the gap. Empirically, they do not.
> > >
> > > ## 3. The deeper contribution is theoretical and conceptual
> > >
> > > More importantly, our contribution is not only a practical combination:
> > >
> > > - **First**, to our knowledge, this is the first work to explicitly connect **contrastive decoding** with **implicit reward** in the W2SG setting. This provides a principled explanation for why CD can expose preference-relevant signal, rather than acting only as a decoding trick.
> > > - **Second**, to our knowledge, this is also the first work to propose transferring the **contrast between weak-model states**, rather than only raw weak-model outputs. This is the central conceptual step of the paper.
> > > - **Third**, this naturally leads to a new W2SG paradigm: using the **improvement-induced weak contrastive signal** to supervise stronger models.
> > >
> > > In this sense, the pipeline is the realization of the idea, not the idea itself. The core novelty is discovering that the **contrast between pre- and post-improved weak models is itself a transferable supervision signal**, and then showing theoretically and empirically that this signal can systematically improve stronger models.
> > >
> > > ---
> > >
> > > In summary, we agree that ConG is implemented with familiar components. However, in the context of W2SG, we respectfully believe the paper contributes:
> > >
> > > - a **new theoretical connection** between CD and implicit reward,
> > > - a **new supervision perspective** for weak-to-strong transfer,
> > > - and a **simple but consistently stronger framework** built on that principle.
> > >
> > > For these reasons, we believe ConG is more than a procedural integration and represents a meaningful conceptual contribution to the W2SG literature. We are sincerely grateful for your thoughtful engagement and positive support, and we hope this clarification helps strengthen your confidence in the novelty and value of the paper.
> > >
> > > Best,
> > >
> > > Authors of Submission 24640
> > >
> > > [1] *Weak-to-Strong Generalization: Eliciting Strong Capabilities With Weak Supervision.* ICML 2024.
> > > [2] *Your Weak LLM is Secretly a Strong Teacher for Alignment.* ICLR 2025.
> > > [3] *How to Mitigate Overfitting in Weak-to-Strong Generalization?* ACL 2025.
> > > [4] *MACPO: Weak-to-Strong Alignment via Multi-Agent Contrastive Preference Optimization.* ICLR 2025.

---

### Official Review · Reviewer_XAE6 · 2026-03-18

**Soundness:** 3
**Presentation:** 3
**Significance:** 2
**Originality:** 3
**Overall Recommendation:** 5
**Confidence:** 3

**Summary:**

This paper studies weak-to-strong generalization for language models. Here, a stronger model is trained using supervision examples generated by a weaker aligned model, with the goal that the stronger model can inherit and extend the weaker model’s capabilities/alignment behavior. The paper’s starting point is that the weak model’s outputs can be noisy or biased, so directly training on those outputs may limit the effectiveness of weak-to-strong transfer.

The paper proposes a method called Contrastive Weak-to-Strong Generalization (ConG). The method uses two versions of the weak model: an unaligned version and an aligned version. The paper argues that the log-likelihood ratio between these two models, which is related to an implicit reward signal, has the same basic structure as contrastive decoding. Motivated by this connection, the method generates responses by contrastively decoding between the unaligned and aligned weak models, and then uses those responses to train the stronger model, first with supervised fine-tuning and then with a preference-optimization step.

The experiments evaluate this framework on instruction-following benchmarks and additional downstream tasks. The reported results suggest that using contrastively generated weak-model responses can improve weak-to-strong transfer relative to more direct ways of using weak-model supervision, while largely preserving the stronger model’s general downstream performance.

**Compliance With Llm Reviewing Policy:**

Affirmed.

**Final Justification:**

as I mentioned in my review, I didn't have any major questions, and, hence, I retain my original assessment after the author's response

**Key Questions For Authors:**

NA

**Limitations:**

Yes

**Strengths And Weaknesses:**

**Strengths**

Weak-to-strong generalization is an active and interesting area, both theoretically and empirically, and alignment of large language models is likewise an important problem. One strength of the paper is that it connects these two directions in a nontrivial way: rather than simply using an aligned weak model to supervise a stronger model, it proposes to contrast an unaligned weak model with its aligned counterpart and use this contrastive signal to train the stronger model. This gives the paper a concrete methodological idea beyond a straightforward application of existing weak-to-strong setups, and it seems like a direction that could potentially be useful more broadly. At the same time, I would have liked the paper to explore these broader possible applications more fully, as I discuss below.

**Weaknesses**

As I understand it, the broader literature on weak-to-strong generalization also considers using weak models to improve or elicit the stronger model’s more general capabilities, rather than only its alignment behavior (e.g., Burns et al., 2024; Mitchell et al., 2024; Somerstep et al., 2025). In comparison, the current paper is primarily focused on improving the alignment of the strong model. While the paper includes additional math/code experiments, these are mainly intended to check that the proposed alignment method does not reduce the strong model’s broader capabilities, rather than to demonstrate that it improves the strong model’s general capabilities. It would be interesting to see whether the idea of contrasting a weak model with a fine-tuned weak model can be extended to settings where the weak model is fine-tuned to improve its reasoning or other capabilities, and whether this can then be used to improve the stronger model more broadly.

In other words, the weakness is the scope of the present paper, which seems narrower than previous weak-to-strong generalization work.

Some typos:
1. Running title throughout is “Training-free Streaming Safeguard with Sparse Autoencoders” whish seems wrong
2. Page 2: “simple yet effect paradigm” should be “simple yet effective paradigm.”

---

> ### Author Rebuttal · Authors · 2026-03-31
>
> Dear Reviewer XAE6:
>
> Thank you for your kind words and positive feedback regarding the novelty, presentation, and effectiveness of our work! Your approval is the great encouragement for us and motivates us to continue advancing our research.
>
> ## *W1: The current paper is mainly focused on improving alignment rather than broader weak-to-strong capability transfer.*
>
> Thank you for this thoughtful comment. We agree that the main paper is more focused on the alignment-oriented weak-to-strong setting. In response:
>
> 1. At the same time, `Appendix D.2` already provides initial evidence that our framework is not limited to instruction-following alignment: **we also evaluate weak-to-strong transfer on math and code domains, where ConG and ConG-S consistently improve over the corresponding base strong models.**
>
> 2. We also understand your broader concern more specifically: whether the same idea can support a single strong model that generalizes across multiple task domains, rather than only improving one domain at a time. To address this, we further conduct an additional **multi-domain weak-to-strong experiment**, where we combine instruction-following, math, and code data into a unified training setting. Concretely, for each domain we first use its corresponding domain-specialized weak model pair to generate CD-based data, and then merge these data to train a single strong model.
>
> The results are summarized below:
>
> |Model|Method|Inst. Avg.|Math Avg.|Code Avg.|
> |:-:|:-:|:-:|:-:|:-:|
> |Qwen2.5-7B|Base|35.2|74.0|77.3|
> ||ConG-S (mixed)|46.8|77.9|79.8|
> ||ConG (mixed)|51.2|79.4|80.7|
> |Llama3-8B|Base|26.5|59.7|65.1|
> ||ConG-S (mixed)|35.6|63.6|67.7|
> ||ConG (mixed)|40.4|64.8|68.6|
>
> These results suggest that the contrastive weak-pair idea can extend beyond alignment-only transfer: after mixing data from multiple domains, ** ConG still improves the same strong model across instruction following, math, and codesimultaneously**. This provides preliminary evidence that our framework may support broader weak-to-strong capability transfer, not only alignment improvement.
>
> We will add this discussion and the new experiment to the revised manuscript to clarify the scope of the method.
>
> ## *W2: Typos and clerical issues.*
>
> Thank you for catching these issues. In response:
>
> 1. The running header was indeed incorrect; this was our oversight, and we have corrected it in the revised version.
> 2. “simple yet effect paradigm” has been corrected to **“simple yet effective paradigm.”**
> 3. During this revision, we also corrected several imprecise or overly strong expressions, including:
>    - **“preserve the full signal of target preferences”** → **“preserve target-preference signals to some extent”**
>    - **“approximately maximize implicit reward”** → CD **biases decoding toward higher implicit-reward responses**
>    - Repeated use of **“equivalence”** → **“correlation”**
>
> We appreciate these careful proofreading comments and have re-checked the manuscript accordingly.
>
> ---
>
> *Once again, we deeply appreciate your thoughtful and encouraging feedback. Your suggestions have not only enhanced the current work but have also inspired us to continue exploring research in the area of W2SG. We are excited to keep moving forward and contributing to the community!*
>
> Best,
>
> Authors of Submission 24640

---

> > ### Author Rebuttal · Reviewer_XAE6 · 2026-04-01
> >
> > Thank you for the rebuttal. My concern is sufficiently addressed. The new multi-domain results are helpful and suggest that the method may have broader applicability beyond the paper’s current alignment-focused framing. I still think a fuller treatment of that broader perspective could make for an even stronger paper, but that would likely require a more substantial rewrite than is necessary here. As is, I think the paper can be accepted in its current framing, and a brief discussion of this broader applicability in the final version would be welcome.

---

> > > ### Author Response · Authors · 2026-04-03
> > >
> > > Dear Reviewer XAE6,
> > >
> > > Thank you very much for your positive feedback and for recognizing that our rebuttal has sufficiently addressed your concern. We truly appreciate your thoughtful assessment and support.
> > >
> > > As you suggested, we will reflect this point more clearly in the final version. In particular:
> > >
> > > - We will briefly discuss that the new multi-domain results suggest our method may extend beyond the current alignment-focused framing.
> > > -  We will add a concise discussion in the final version to highlight this broader applicability while keeping the current framing of the paper.
> > >
> > > Thank you again for your thoughtful comments and support throughout the review process.
> > >
> > > Best regards,
> > > Authors of paper 24640

---

### Decision · Program_Chairs · 2026-04-30

**Decision:**

Accept (regular)

**Comment:**

This paper introduces Contrastive Weak-to-Strong Generalization (ConG), a framework that enhances W2SG by leveraging contrastive decoding to generate denoised, high-quality supervision for training stronger models.
Some reviewers view the method as a "combinatorial integration" of existing techniques, but the paper justifies its approach by demonstrating the structural consistency between contrastive decoding and implicit rewards, which provides insights into how these techniques can be effectively applied to improve W2SG. Additionally, the paper provides additional empirical evidence in response to these concerns regarding the over-reliance on GPT-4 for evaluation and the potential impact on the strong model's broader capabilities.
Overall, while the paper could benefit from a fuller treatment of its broader applications, the empirical results and theoretical foundation make it a solid contribution. Therefore, I am inclined to recommend weak accept for this paper.